# Design, Synthesis, and Structure–Activity Relationship Studies of New Quinone Derivatives as Antibacterial Agents

**DOI:** 10.3390/antibiotics12061065

**Published:** 2023-06-16

**Authors:** Juan Andrades-Lagos, Javier Campanini-Salinas, América Pedreros-Riquelme, Jaime Mella, Duane Choquesillo-Lazarte, P. P. Zamora, Hernán Pessoa-Mahana, Ian Burbulis, David Vásquez-Velásquez

**Affiliations:** 1Facultad de Medicina y Ciencia, Universidad San Sebastián, Santiago 7510157, Chile; 2Drug Development Laboratory, Faculty of Chemical and Pharmaceutical, Sciences, Universidad de Chile, Santiago 8380492, Chile; 3Facultad de Medicina y Ciencia, Universidad San Sebastián, Puerto Montt 5501842, Chile; 4Instituto de Química y Bioquímica, Facultad de Ciencias, Universidad de Valparaíso, Playa Ancha, Valparaíso 2360102, Chile; 5Centro de Investigación Farmacopea Chilena (CIFAR), Facultad de Farmacia, Universidad de Valparaíso, Playa Ancha, Valparaíso 2360102, Chile; 6Laboratorio de Estudios Cristalográficos, IACT (CSIC-UGR), Av. de las Palmeras 4, 18100 Armilla, Spain; 7Departamento de Química y Biología, Facultad de Ciencias Naturales, Universidad de Atacama, Copiapó 1530000, Chile; 8Departamento de Química Orgánica y Fisicoquímica, Facultad de Ciencias Químicas y Farmacéuticas, Universidad de Chile, Santiago 8380492, Chile; 9Centro de Investigación Biomédica, Facultad de Medicina y Ciencias, Universidad San Sebastián, Sede de la Patagonia, Puerto Montt 5501842, Chile

**Keywords:** synthesis, antibacterial agents, quinonic-antibiotics, structure–activity relationships, Craig plot, methicillin-resistant *Staphylococcus aureus*, *Enterococcus faecium*, *Klebsiella pneumoniae*, antibacterial activity, drug discovery, quinone-antibiotics, Free-Wilson

## Abstract

Resistance to antibacterial agents is a growing global public health problem that reduces the efficacy of available antibacterial agents, leading to increased patient mortality and morbidity. Unfortunately, only 16 antibacterial drugs have been approved by the FDA in the last 10 years, so it is necessary to develop new agents with novel chemical structures and/or mechanisms of action. In response to this, our group takes up the challenge of designing a new family of pyrimidoisoquinolinquinones displaying antimicrobial activities against multidrug-resistant Gram-positive bacteria. Accordingly, the objective of this study was to establish the necessary structural requirements to obtain compounds with high antibacterial activity, along with the parameters controlling antibacterial activity. To achieve this goal, we designed a family of compounds using different strategies for drug design. Forty structural candidates were synthesized and characterized, and antibacterial assays were carried out against high-priority bacterial pathogens. A variety of structural properties were modified, such as hydrophobicity and chain length of functional groups attached to specific carbon positions of the quinone core. All the synthesized compounds inhibited Gram-positive pathogens in concentrations ranging from 0.5 to 64 µg/mL. Two derivatives exhibited minimum inhibitory concentrations of 64 µg/mL against *Klebsiella pneumoniae*, while compound 28 demonstrated higher potency against MRSA than vancomycin.

## 1. Introduction

Resistance to antibacterial agents is a global problem that often eliminates treatment options in remote locations for several infectious diseases to directly increase mortality [1]. There will be approximately 10 million increased deaths caused by resistant microorganisms by 2050 if no initiatives solve this problem [2]. Investment in creating new antimicrobials has steadily decreased over the last ten years despite the alarming increase in drug resistance [3]. The development of compounds with new mechanisms of action has also decreased in the last decade [3,4]. This problem is worsened by a general lack of interest from the pharmaceutical industry in this market [5,6]. The insufficient investment can be attributed to the perceived low economic return for these types of drugs compared to other pharmacological targets, including treatments for chronic diseases [3,6]. Moreover, this lack of interest is further compounded by various factors. These factors include the focus on short-term curative treatments [3,6], the strict control and restricted use of new antibiotics [7,8], the emergence of generic forms after the expiration of intellectual property patents [6], increased demands by the FDA to demonstrate the efficacy of new antibacterial agents [7], and lastly, the occurrence of drug resistance prior to or shortly after their introduction to the market. Each of these reasons contributes to a decrease in their use and expected economic return [5]. This sentiment is reflected in the fact that only 16 antibacterial compounds have been introduced to the market since 2013 [9,10,11].

To address this problem, public and non-governmental organizations (NGOs) have started more than 50 initiatives to develop new antibacterial drugs based on known compounds that would likely never be developed in the private pharmaceutical sector. These public-private initiatives include the Joint Programming Initiative on Antimicrobial Resistance (JPIAMR), the Innovative Medicines Initiative’s (IMI’s) New Drugs for Bad Bugs (ND4BB) Program, the Biomedical Advanced Research and Development Authority’s (BARDA) Broad Spectrum Antimicrobials Program, and the Combating Antibiotic Resistant Bacteria Biopharmaceutical Accelerator (CARB-X) [12]. The World Health Organization (WHO) published a list of priority pathogens to guide R&D efforts in the development of new antibacterial drugs called [10].

Within the critical priority group are bacteria such as *Pseudomonas aeruginosa* and enterobacteria (such as *Escherichia coli* and *Klebsiella* sp.) resistant to carbapenems and third-generation cephalosporins. On the other hand, bacteria such as vancomycin-resistant *Enterococcus faecium* and methicillin- and vancomycin-resistant *Staphylococcus aureus* are still under-served and remain classified as high-priority pathogens [10].

In this context, we previously described the synthesis and evaluation of a collection of arylmercaptoquinonic derivatives that exhibit activity against vancomycin-resistant *Enterococcus faecium* (VREF) and methicillin-resistant *Staphylococcus aureus*. These compounds demonstrated 128-fold higher activity against clinical isolates of VREF compared to vancomycin while not affecting the viability of HeLa, HTC-116, SH-SY5Y, or Vero cells in toxicity assays [13,14]. However, there is still a significant gap in our understanding regarding how modifications to the quinone core can impact their biological activity. Based on this, we designed an extensive series of new pyrimidoisoquinolinquinone derivatives and synthesized them to test their antibacterial activities against high-priority pathogens declared by the WHO. The objective of this study was to investigate the structure–activity relationship of this novel family of antibiotic compounds.

To explore this possibility, we designed an extensive series of new pyrimidoisoquinolinquinone derivatives, carrying out their synthesis and testing antibacterial activities using high-priority pathogens declared by the WHO.

The structural requirements necessary to obtain compounds with high antibacterial activity were identified together with the defined parameters that modulated antibacterial profiles. Computational chemistry and crystallographic studies were incorporated to explain the obtained results. The structure–function relationships of these compounds were explored in the context of developing lead drugs for further investigation.

## 2. Results and Discussion

### 2.1. Drug Design

Recently, we reported a new kind of quinone-antibiotic exhibiting anti-infective properties against different Gram-positive pathogens [13]. In this study, only a few modifications on the thiophenolic ring substituent were carried out; this reason led us to develop extents on the study of the structure–activity relationship considering a rational design, synthesis, and evaluation of the antibacterial activity of novel compounds of general structure pyrimidoisoquinolinquinone. The **P1** structure (Table 1) was selected as a prototype for further optimization using five optimization strategies. First, the Craig model allowed us to analyze the influence of para-aromatic substituents on biological activity. To contextualize the chemical activities of the derivatives, we grouped theoretical structures in a cartesian graph whose X and Y axes corresponded to the lipophilicity (π) and Hammett substituent constant (σ) parameters of the substituents. For the development of this strategy, we used the compounds previously synthesized in Campanini et al. work [13]. We created a theoretical derivative space in which all possible substituent combinations in the four quadrants of π and σ were displayed in the para position. We calculated molar refractivity (MR) to evaluate the steric influence of substituents on antibacterial activity. Next, we tested classic isosteric replacement by substituting the sulfur atom with nitrogen to determine whether another heteroatom could be used in this position. Third, we performed double substitutions on the thiophenolic ring using a Free-Wilson analysis as a tool to evaluate if the disubstituted compounds had antibacterial activity. Fourth, we performed three homology substitution studies to explore: (1) the effect of the distance between the aromatic ring and the sulfur atom, (2) the effect of the addition of thioalkyl derivatives instead of thioaryl substitutions with the purpose of determining whether the compounds with alkylic chain possess antibacterial activity and the influence of its carbon chain length on the activity, and (3) the effect of modifying position 6 of the tricyclic quinone core on overall activity. We finally added a second chemical group to the tricyclic quinone core to create homodisubstituted derivatives in positions 8 and 9.

A summary of the modifications performed is shown in Figure 1.

### 2.2. Synthesis

The compounds were synthesized in two consecutive stages. In the first stage, the tricyclic quinone cores **3**, **4**, and **5** were obtained according to the general procedure A, described by Valderrama et al. [15] and Campanini et al. [13]. This step proceeds via a ‘one-pot’ reaction, starting with the oxidation of the respective hydroquinone precursor to the quinone ring with silver oxide I at room temperature. The activated quinone reacted with the aminouracil via [3 + 3] cyclization giving rise to the tricyclic hydroquinone intermediate, which was rapidly oxidized aerobically to the respective quinone core. The proposed reaction mechanism is shown in Figure 2.

In the second stage, regioselective addition to the quinone is performed using four different procedures based on the reactivity of the precursors. For compounds with aniline derivatives, the direct addition of the complete reagent, in equimolar amounts relative to the tricyclic quinone, to the reaction mixture resulted in the predominant formation of only the C-8 regioisomer, corresponding to the derivative with the nitrogen atom attached to the 8-position of the tricyclic quinone core. In this way, the addition of higher proportions of reagent to quinone (2:1) did not generate diaddition products (substituted in 8- and 9-positions). The C-9 regioisomer was not detected in any case. On the other hand, compounds with thiophenol and alkylthio derivatives were slowly added dropwise to the reaction mixture in order to avoid the formation of diaddition compounds. Through this procedure, it was observed that the major products corresponded to the C-8 regioisomers. Similar to the compounds with aniline derivatives, the formation of the C-9 regioisomer was not observed. Finally, considering the possibility of diaddition with thiophenol or alkylthio derivatives, a new procedure (“D”) was developed to obtain compounds C-8 and C-9 substitution in order to include this type of structure in the analysis of structure–activity relationship. A summary of the general procedures for obtaining the compounds is shown in Figure 3, and the Appendix A.

The regioselectivity observed in these reactions with quinones **3** and **4** can be explained by the effect of the cerium ion, which acts as a catalyst [16] that favors the nucleophilic attack at position C-8 by coordination with the nitrogen heteroatom and/or with the carbonyl group at position C-10, favoring its electron-attractor character and thus allowing the nucleophilic substitution at C-8 [17]. With these results, general procedure B was employed for the thiophenolic compounds **7**–**9**, **11**–**21**, **30**–**34,** and **38**–**47,** and general procedure C for the arylamines compounds **22**, **24,** and **26**–**29**. Given the synthetic possibility of obtaining homodisubstituted compounds via aerobic oxidation, the general procedure D was developed, affording compounds **35**–**37**.

Interestingly, the synthetic route described in general procedure B for the tricyclic quinonic core **5** produced a mixture of C-8 and C-9 isomers in a 73:27 ratio. These results were similar to those reported by Valderrama et al., who studied the addition of cycloalkyl-, n-alkyl-, and arylamines to the phenanthridin-7,10-quinone core [18].

Additionally, compound **11** was subjected to an aryl nitro reduction reaction with iron powder in an acidic medium which allowed obtaining compound **6** [18]. Due to the reactivity of the precursors, some modifications to the corresponding general procedures were necessary to obtain compounds **10**, **23,** and **25**.

### 2.3. Antibacterial Activity

Compounds target were tested in vitro against *Staphylococcus aureus* methicillin-susceptible strain (ATCC^®^ 29213), *Staphylococcus aureus* methicillin-resistant strain (ATCC^®^ 43300), *Enterococcus faecalis* (ATCC^®^ 29212), *Escherichia coli* (ATCC^®^ 25922), *Pseudomonas aeruginosa* (ATCC^®^ 27853), and *Klebsiella pneumoniae* (ATCC^®^ 700603) by minimum inhibitory concentration (MIC) using microbroth dilutions technique using Müeller–Hinton broth, according to recommendations of the Clinical and Laboratory Standards Institute (CLSI) [13].

The screening results of these new compounds for antibacterial activity in vitro are reported in Table 1. Previously synthesized compounds are designated by the letter P.

**Table 1 antibiotics-12-01065-t001:** Antibacterial activities of compounds **3**–**43**, **P1**–**P7,** and antibiotics controls.

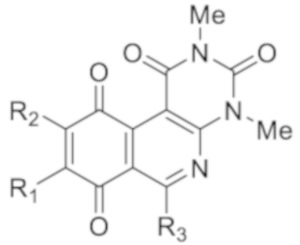										
Compounds					MIC (µg/mL)					
Label	SP ^a^	R_1_	R_2_	R_3_	MSSA (ATCC 29213)	MRSA (ATCC 43300)	*E. faecalis* (ATCC 29212)	*E. coli* (ATCC 25922)	*P. aeruginosa* (ATCC 27853)	*K. pneumoniae* (ATCC 700603)
**3**	**A**	H	H	Et	>32	>32	>32	>32	>32	>32
**4**	**A**	H	H	Me	>32	>32	>32	>32	>32	>32
**5**	**A**	H	H	H	>32	>32	>32	>32	>32	>32
**6**	**-** ^b^	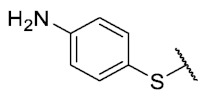	H	Et	>32	>32	32	>32	>32	>32
**7**	**B**	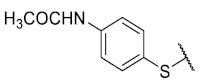	H	Et	16	16	16	>32	>32	>32
**8**	**B**	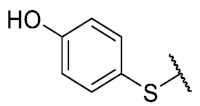	H	Et	16	16	16	>32	>32	>32
**9**	**B**	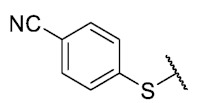	H	Et	8	8	8	>32	>32	>32
**10**	**-** ^b^	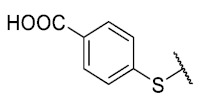	H	Et	32	16	32	>32	>32	>32
**11**	**B**	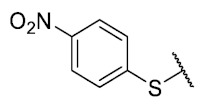	H	Et	8	8	8	>32	>32	64
**12**	**B**	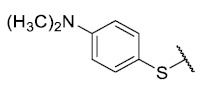	H	Et	16	8	16	>32	>32	>32
**13**	**B**	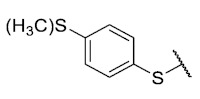	H	Et	>32	>32	>32	>32	>32	>32
**14**	**B**	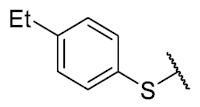	H	Et	4	4	8	>32	>32	>32
**15**	**B**	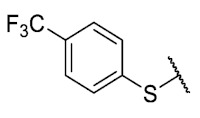	H	Et	>32	4	>32	>32	>32	>32
**16**	**B**	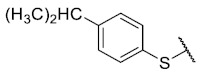	H	Et	>32	>32	>32	>32	>32	>32
**17**	**B**	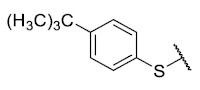	H	Et	>32	>32	>32	>32	>32	>32
**18**	**B**	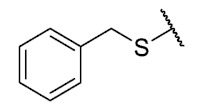	H	Et	>64	>64	64	>64	>64	>64
**19**	**B**	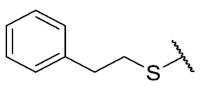	H	Et	>32	>32	>32	>32	>32	>32
**20**	**B**	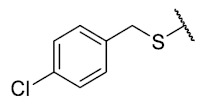	H	Et	>32	>32	>32	>32	>32	>32
**21**	**B**	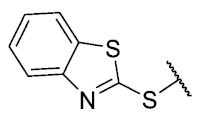	H	Et	>32	>32	>32	>32	>32	>32
**22**	**C**	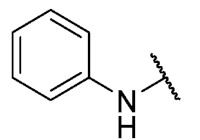	H	Et	>32	>32	>32	>32	>32	>32
**23**	**-** ^b^	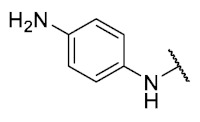	H	Et	>32	>32	>32	>32	>32	>32
**24**	**C**	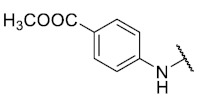	H	Et	>16	>16	>16	>16	>16	>16
**25**	**-** ^b^	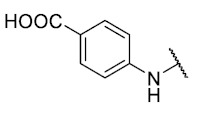	H	Et	>32	>32	>32	>32	>32	>32
**26**	**C**	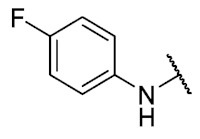	H	Et	>32	>32	>32	>32	>32	>32
**27**	**C**	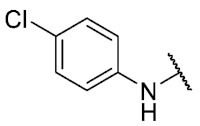	H	Et	1	1	4	>32	>32	>32
**28**	**C**	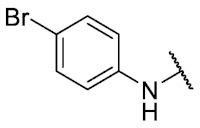	H	Et	0.5	0.5	4	>32	>32	>32
**29**	**C**	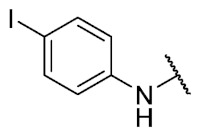	H	Et	>32	>32	>32	>32	>32	>32
**30**	**B**	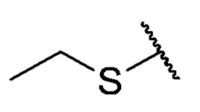	H	Et	>8	>8	>8	>8	>8	>8
**31**	**B**	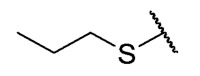	H	Et	8	4	8	>32	>32	>32
**32**	**B**	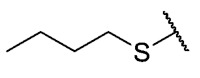	H	Et	4	4	4	>32	>32	>32
**33**	**B**	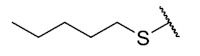	H	Et	>16	>16	>16	>16	>16	>16
**34**	**B**	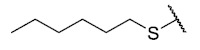	H	Et	>16	>16	>16	>16	>16	>16
**35**	**D**	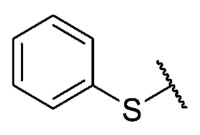	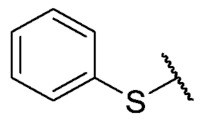	Et	>16	>16	>16	>16	>16	>16
**36**	**D**	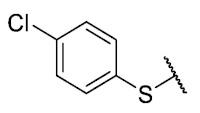	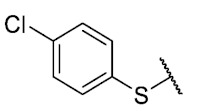	Et	>16	>16	>16	>16	>16	>16
**37**	**D**	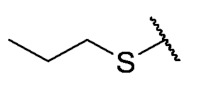	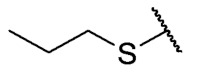	Et	>32	>32	>32	>32	>32	>32
**38**	**B**	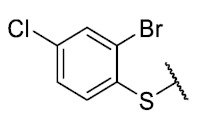	H	Et	>4	4	>4	>4	>4	NT
**39**	**B**	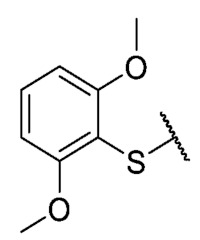	H	Et	2	2	2	>64	>64	>64
**40**	**B**	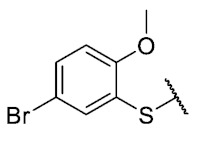	H	Et	>32	>32	>32	>32	>32	>32
**41**	**B**	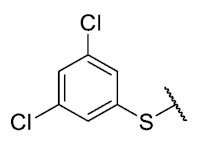	H	Et	>64	4	32	>64	>64	64
**42**	**B**	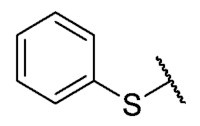	H	Me	8	8	8	>32	>32	>32
**43**	**B**	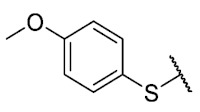	H	Me	8	8	8	>32	>32	>32
**44**	**B**	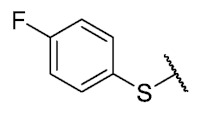	H	Me	4	8	8	>32	>32	>32
**P1**		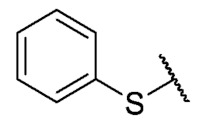	H	Et	8	8	8	>32	>32	NT
**P2**		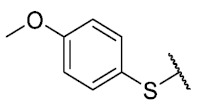	H	Et	16	16	16	>32	>32	NT
**P3**		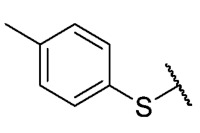	H	Et	4	4	16	>32	>32	NT
**P4**		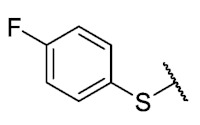	H	Et	8	8	8	>32	>32	NT
**P5**		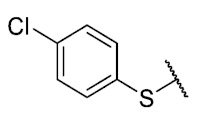	H	Et	4	4	4	>32	>32	NT
**P6**		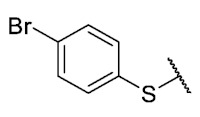	H	Et	4	8	8	>32	>32	NT
**P7**		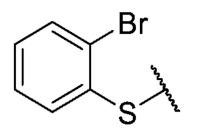	H	Et	4	1	2	>32	>32	NT
VAN **^c^**					1	1	2	NT	NT	NT
GEN **^d^**					NT	NT	NT	0.5	1	2

^a^ SP = Structure Procedure. ^b^ See section of structural procedures. ^c^ vancomycin, quality control for Gram-positive ATCC^®^ strains is 0.5–2 µg/mL against MRSA and MSSA; 1–4 µg/mL against *E. faecalis* according to CLSI [19]. ^d^ gentamicin, quality control for Gram-negative ATCC^®^ strains are 0.25–1 µg/mL against *E. coli* and 0.25–2 against *P. aeruginosa* according to CLSI [19]. NT = Not tested; MIC = Minimum inhibitory concentration; VAN = vancomycin; GEN = gentamicin.

The results show that the compounds present antibacterial activity on Gram-positive bacteria within the range of 0.5 to 64 µg/mL. The most active compounds of the series correspond to molecules **28**, **27**, **39**, **32,** and **14,** with activities 0.5, 1, 2, 4, and 4 µg/mL, respectively, for MSSA and MRSA. On the other hand, for *E. faecalis*, the same molecules presented activities of 4, 4, 2, 4, and 8 µg/mL, respectively. On the other hand, no activity was observed against Gram-negative bacteria except for compounds **11** and **41** which were active against *K. pneumoniae* ATCC^®^ 700603 at a concentration of 64 µg/mL.

A detailed study of the structure–activity relationship of the compounds is presented as follows.

## 3. Structure–Activity Relationship

### 3.1. Craig Model

For the development of the Craig model [20], p-substituted compounds **6**–**17** were used, including the previously reported compounds **P1**–**P6**.

First, p-substituted target compounds were designed using substituents from the four quadrants of a Craig plot to explore the electronic and hydrophobic space around the benzylthio moiety and their effect on activity. We added previously synthesized compounds to this list to expand the landscape to compare these new derivatives. Our results showed that the lipophilic character of the substituent in the para position increases the antibacterial activity of these compounds against Gram-positive bacteria with two outlier compounds (**10** and **13**). These data establish a maximum value of lipophilicity for the substituents in compounds **14** (4-ethyl) and **15** (4-CF_3_). The compounds with the most lipophilic substituents, **16** (4-iso-propyl) and **17** (4-tert-butyl), did not exhibit antibacterial activity at the concentrations tested (>32 µg/mL).

We observed that the electronic character (Hammett substituent constant) of the substituent had a minimal influence on the antibacterial activity. For example, when comparing compounds **6** (4-NH_2_) and **12** (4-N(CH_3_)_2_), which present similar values of σ and different values of π present MIC values of >32 and 8 µg/mL, respectively, in MRSA. On the other hand, compounds such as **P1** (4-H), **P4** (4-F), and **11** (4-NO_2_), which have similar π values, have the same MIC value (8 µg/mL) in all the Gram-positive bacteria studied. Additionally, the volume of the substituent, calculated through the molar refractivity parameter, does not show a direct relationship with the antibacterial activity of these compounds. Thus, the lipophilic p-thiophenolic substitution would favor the antibacterial activity of this series of a compound.

### 3.2. Bioisosteric Replacement

For the bioisosteric replacement analysis, the antibacterial activity of compounds bearing a sulfur atom **P1**, **6**, **10**, and **P4**–**P6** were compared with the compounds having a nitrogen atom in **22**, **23**, **25**–**28**, respectively.

The replacement of the sulfur atom by nitrogen (**22**–**28**) caused the loss of activity for compounds **22**–**26**. In contrast, derivatives **27** (*p*-Cl aniline) and **28** (*p*-Br aniline) showed activity in Gram-positive bacteria in a range of 0.5–4 μg/mL, establishing themselves as the most active compounds of the entire series. It is observed that the aniline derivatives (**27**–**28**) are between 2 and 16 times more potent than their thiophenolic analogs, **P5** and **P6**, respectively. In addition, the antibacterial activity of compound **28** is twice that of vancomycin (0.5 μg/mL vs. 1 μg/mL) in MSSA and MRSA.

To study the effect of bioisosteric replacement on the geometry of the compounds, the compounds with nitrogen atoms **23**, **27,** and **28** were crystallized and resolved by X-ray diffraction to compare them with the compounds with sulfur atoms **P5** and **P6**. The data for compound **P5** were extracted from the study by Campanini et al. [13]. (Figure 4).

The three-dimensional models showed that the dihedral angle of the sulfur derivatives is 76.4° and 72.3° for the compounds **P5** and **P6,** respectively, while for the compounds with nitrogen, they correspond to 31.9° and 32.3°, respectively. Since compounds **27** and **28** are at least four times more active than their bioisosteres **P5** and **P6**, this result suggests that the activity improves when both aromatic systems are more coplanar, probably due to the conjugation between the benzene ring and the aromatic system aromatic of the quinone core. In addition, it is interesting to note that within the series of compounds with a nitrogen atom, the presence of an electron-attracting group, as is the case of the chlorine and bromine atoms in the 4′ position of the benzene ring, since a monovalent bioisosteric replacement by an amino group (compound **23**), maintains the dihedral angle (38.6°) close to compounds **27** and **28** but eliminates the activity completely.

In addition, considering the results of the Craig plot and the bioisosteric change, it was decided to synthesize the derivative 6-ethyl-8-((4-iodophenyl)amino)-2,4-dimethylpyrimido[4,5-*c*]isoquinolin-1,3,7,10(2*H*,4*H*)-tetraone (compound **29**). This compound did not present activity in the evaluated bacteria (MIC > 32 µg/mL), which can be explained by the fact that the hydrophobicity value of iodine is higher than that of the ethyl and CF_3_ groups, established as the substituents with the highest hydrophobicity allowed for the series of thiophenolic compounds.

Thus, bioisosteric replacement allowed the generation of the two most active compounds of the series against Gram-positive bacteria, presenting in vitro antibacterial activity similar to vancomycin against MRSA.

### 3.3. Free-Wilson Study

Considering the clinical relevance of MRSA infections, this antibacterial data was selected for use in the Free-Wilson statistical analysis to understand the contribution of each substituent in the aromatic ring with the aim of generating new active compounds. A Free-Wilson analysis is a numerical method that directly relates structural features with biological properties [21]. In our case, we have correlated the presence or absence of the different substituents in *ortho*, *meta*, and *para* positions on the benzene ring with the biological activity of every compound expressed as pMIC (−logMIC, in Molar units). For this purpose, we constructed a matrix in which the presence of a substituent (Me, MeO, F, Cl, or Br) is represented by 1 and absent a 0. Then, through a multilinear regression, we correlated the biological activity (dependent variable) with the matrix of 1 and 0. The results of the Free-Wilson analysis are shown in Table 2.

The coefficient will be positive for the presence of a substituent that improves the biological activity when it is placed in that position and is the opposite when the coefficient is negative. As can be seen in Table 2, the worst substituents are Me in the *ortho*, and MeO in the *para* position; therefore, such substituents should be avoided to improve anti-MRSA activity. On the other hand, the best substituents are MeO in the ortho position*,* Cl and Br in the *meta* position, and Br in the para position. This could be explained by both the increase in the lipophilicity of the compounds and by the capacity of these halogens to establish electrostatic interactions with their target.

Based on these results, the following compounds were synthesized to challenge the model Table 3.

The antibacterial activity results showed that compounds **38**, **39,** and **41** exhibited antibacterial activities of 4, 2, and 4 µg/mL against MRSA. In addition, compound **39** exhibited activity against MSSA and *E. faecalis* at 2 µg/mL, and compound **41** was active against *E. faecalis* at 32 µg/mL. This compound stands out for presenting activity against *K. pneumoniae* at 64 µg/mL. Compound **40** did not exhibit activity against the bacteria tested (>32 µg/mL). To explore the structural aspects of these derivatives, we crystallized compound **38** and compared it with compounds **P5** and **P7** reported in the study by Campanini et al. [13] (Figure 5A).

We observed that the dihedral angle of compound **38** is 70.6°, minor than **P5** and **P7**, which correspond to 76.4° and 81.5°. Due to the low solubility of these compounds, it is possible to only observe antimicrobial activity in MRSA with a value of 4 µg/mL, which is similar for **P5** but low for **P7**, which presents 4 and 1 µg/mL, respectively. These results indicated that it is possible to perform a second substitution on the thiophenolic ring to obtain active compounds.

### 3.4. Homology Study

Three chain extensions were performed in different sections of the structure to understand its relationship with activity. In the first scenario, the compounds **P1** (p-H) and **P5** (p-Cl) [13] were used as references for studying the effect of the distance between the phenyl ring and the sulfur atom on the antibacterial activity. The comparison of **P1** with **18** shows that homologation by one methylene group decreased the antibacterial activity of de 8 µg/mL (**P1**) a 64 µg/mL (**18**) in *E. faecalis* and produced the loss of activity (>64 µg/mL) in MRSA and MSSA. When the distance was extended by inserting a second methylene group, the solubility decreased (**18** vs. **19**) and resulted in the complete loss of antibacterial activity (>64 µg/mL). For the substituted p-Cl pair **P5** and **20**, the loss of antibacterial activity from 4 µg/mL (**P5**) to >32 µg/mL (**20**) is reproducible when a methylene group is added between the thioether and the aromatic ring. These results suggest that extending the distance between the thioether and the aromatic ring and thus increasing the degrees of freedom of the molecule has a deleterious effect on antibacterial activity in the three Gram-positive bacteria evaluated, possibly due to the loss of conjugation between the aromatic system and quinonic nucleus.. The addition of a benzo[d]thiazole-2-thiol derivative was performed to determine the effect on the biological activity if increasing the size and rigidity of the system was associated with a change. In this way, we synthesized compound **21**, which did not exhibit antibacterial activity (>32 µg/mL) in any of the bacteria tested. Thus, the optimal antibacterial activity was obtained when there were no carbon atoms between the thiophenolic ring and the tricyclic quinone core.

The second homology study sought to evaluate the addition of thioalkyl chains from 2 to 6 carbon atoms (**30**–**34**) to core 3 as an alternative to thiophenolic derivatives. This gave rise to two active compounds, derivatives **31** (3C) and **32** (4C), which presented antibacterial activity in Gram-positive bacteria with MIC values between 4 and 8 μg/mL, establishing the ideal number of carbons as 4. The compounds with fewer carbon atoms (**30**) or more (**33** and **34**) did not present antibacterial activity in any of the bacteria evaluated.

Chemical models obtained by crystallography showed that compounds **30** and **32** do not adopt differences in the structural geometry of the quinone nucleus or in the alkyl chain (Figure 5B). This evidence indicated that the activity exhibited by **32** is due mainly to the increase in lipophilicity of the molecule. It is important to note that alkyl substitutions at position 8 of the tricyclic quinone core can be considered only if their length is 3 or 4 carbons. Other chain lengths generate inactive compounds.

Finally, the impact of the alkyl chain at position 6 of the tricyclic quinone core was evaluated. In such a sense, compounds **42**–**44** (Series 6-Me) were synthesized from compound **4**. When comparing the 6-Et vs. 6-Me series, no differences in antimicrobial activity were observed, showing that shortening of the alkyl radical a methylene group does not affect activity. To extend the analysis and consider the obtaining of the tricyclic quinone core **5**, procedure B was carried out in an exploratory manner to obtain the 6-H series, with the thiophenol, 4-methoxy-thiophenol, and 4-fluoro-thiophenol substituents. NMR analysis showed the presence of a mixture of isomers of C-8 y C-9 in a 73:27 ratio (data not shown). When evaluating the activity of these isomer mixtures, we observed antibacterial activity (data not shown). These findings open the possibility of characterizing the antimicrobial activity of the different isomers, which until now has not been possible due to the regioselective addition of the substituents.

### 3.5. Homodisubstitution Study

The addition of a second substitution in the quinone tricyclic core gives rise to the compounds called homodisubstituted, whose positions 8 and 9 are occupied. From derivatives **P1**, **P5,** and **31**, compounds **35**, **36,** and **37**, respectively, were synthesized. These results indicated that homodisubstitution resulted in the loss of antibacterial activity.

## 4. Conclusions

We used drug optimization tools to establish the structure–activity relationship of this new family of 2,4-dimethylpyrimido[4,5-*c*]isoquinoline-1,3,7,10(2*H*,4*H*)-tetraone derivatives with antimicrobial activity. Nineteen compounds exhibited antibacterial activity against ATCC^®^ Gram-positive bacteria in a range of 0.5 to 64 µg/mL, with compound **28** being the most active of the series. Compound **28** was twice as potent as vancomycin on MRSA. In addition, compounds **11** and **41** showed activity against *K. pneumoniae* ATCC^®^ 700603 at a concentration of 64 µg/mL. This work supports the conclusion that rational drug design can provide useful insights to guide derivative synthesis that accelerates new drug discovery with potentially novel targets of bioactivity.

For compounds with the bridging sulfur atom bearing a para-substituted benzene ring, the antimicrobial activity is favored when the lipophilicity of the molecule increases. A second substitution on the benzene ring generates active compounds; however, these compounds do not show greater activity than the monosubstituted derivatives. Both the introduction of carbons between the sulfur atom and the benzene ring, as well as the replacement of the aromatic ring by an alkyl chain, annuls the activity of the compounds. The replacement of an ethyl group by a methyl group at the C-6 position of the quinone tricyclic core generates active compounds with similar activity. On the other hand, the elimination of the group in position 6 modifies the reactivity of the quinone tricyclic core, generating a mixture of isomers exhibiting less antimicrobial activity. Homo disubstitution of thiophenolic or thioalkyl groups on the quinone core generated inactive compounds.

In addition, the bioisosteric replacement of the sulfur atom by one of nitrogen produced a change in the geometry by reducing the dihedral angle between the substituted benzene ring and the quinone nucleus. This change increased the activity when the benzene ring presented an electro-attracting atom such as chlorine or bromine; on the other hand, the presence of electron donor groups such as amines lowered the activity.

These results showed that this new family of compounds displayed a high potential for improvement in their performance as potent antimicrobial drugs against Gram-positive bacteria. Finally, it is important to highlight the bioactivity of compounds 11 and 41, which open the way for studying new active structures on *K. pneumoniae*, a pathogen considered a priority by the WHO.

## 5. Materials and Methods

### 5.1. Materials

All reagents were purchased from AK-scientific, Union City, United States; Enamine, Kyiv, Ukraine; Merck, Darmstadt, Germany; or Sigma–Aldrich, Burlington, United States and were used without purification. Melting points (mp) were determined on a Stuart Scientific SMP3 apparatus and were uncorrected. ^1^H-NMR spectra were recorded on Bruker AM-400 instruments in deuterochloroform (CDCl_3_) or dimethylsulfoxide (DMSO-d_6_). ^13^C-NMR spectra were obtained in CDCl_3_ or DMSO-d_6_ at 100 MHz. The assignments of chemical shifts are expressed in ppm downfield relative to tetramethylsilane (TMS, δ scale), and the coupling constants (*J*) are reported in Hertz. Silica gel (70–230 and 230–400 mesh) and TLC aluminum foil 60 F254 (Merck, Darmstadt, Germany) were used for preparative column chromatography and analytical TLC, respectively. High-resolution mass spectra were obtained on a mass spectrometer with flight time analyzer (TOF) and Triwave^®^ system model SYNAPT™ G2 (WATERS, Milford, MA, USA), using atmospheric pressure ionization with electro spray (ESI+/−), Capillarity 3.0, source temperature 100 °C, desolvation temperature 500 °C.

### 5.2. Chemical Synthesis

#### 5.2.1. General Procedure A for the Synthesis of **3**–**5**

A suspension of hydroquinone precursor corresponding (**1a–1c**) (1.90 mmol), 6-amino-1,3-dimethyl-2,4(1*H*,3*H*)-pyrimidinedione (**2**) (1.90 mmol), Ag_2_O (5.7 mmol) and anhydrous MgSO_4_ (5.7 mmol), in 40 mL of dichloromethane was stirred vigorously at room temperature for 3 h. The mixture was filtered with celite and washed with dichloromethane. The solvent was removed under reduced pressure, and the crude reaction was purified using 65 g of silica gel (230–400 mesh) using a mix of dichloromethane and ethyl acetate = 9:1 as eluent. The resulting solution was concentrated to dryness under reduced pressure. The obtained products were yellow solids for all tricyclic quinone cores.

#### 5.2.2. General Procedure B for Synthesis of 8-Thioaryl (thioalquil)-pyrimidoisoquinolinequinones Derivatives (**7**–**9**, **11**–**21**, **30**–**34**, **38**–**47**)

A solution of 3 (150 mg, 0.4909 mmol 1.0 equiv.) and CeCl_3_∙7H_2_O (5% mmol respect to 3) in a mix of ethanol: dichloromethane = 1:1 (10 mL) was added dropwise slowly a solution of benzenethiol or alquilthiol derivate (0.5 equiv.) in ethanol: dichloromethane = 1:1 (30 mL). The reaction mixture was stirred at room temperature for 16 h. The progress of the reaction was followed by thin-layer chromatography (TLC). The reaction mixture was concentrated under reduced pressure, and the crude of the reaction was purified using 65 g of silica gel (70–230 mesh) and a mix of dichloromethane, light petroleum, and ethyl acetate as an eluent in determinate proportions. The resulting solution was concentrated to dryness under reduced pressure.

#### 5.2.3. General Procedure C for Synthesis of 8-Arylamino-pyrimidoisoquinolinequinones Derivatives (**22**, **24**, **26**–**29**)

A solution of **3** (150 mg, 0.4909 mmol 1.0 equiv.), the required amine (2 equiv.), CeCl_3_∙7H_2_O (5% mmol respect to **3**), in a mix of ethanol: dichloromethane = 1:1 (10 mL), was left with stirring at room temperature until completion of the reaction indicated by thin-layer chromatography (TLC). The reaction mixture was concentrated under reduced pressure. The crude reaction was purified by column chromatography using 65 g of silica gel (70–230 mesh) and a mix of dichloromethane, chloroform, light petroleum and/or ethyl acetate as eluent in different proportions. The resulting solution was concentrated to dryness under reduced pressure.

#### 5.2.4. General Procedure D for Synthesis of 8,9-Bisthioaryl (Thioalquil)-pyrimidoisoquinolinequinones Derivatives (**35**–**37**)

The bis-substituted derivatives were achieved by preparing a solution of **3** (150 mg, 0.4909 mmol 1.0 equiv.), the corresponding thiophenol derivative (2.5 equiv.), and CeCl_3_∙7H_2_O (5% mol of **3**) in ethanol (40 mL) under reflux conditions for 1–4 h, the progress of the reaction was followed by thin-layer chromatography (TLC). Once the reaction was over, the mixture was cooled to room temperature and stirred under aerobic conditions for 18 h. The reaction mixture was concentrated under reduced pressure, and the crude was purified by column chromatography using 65 g of silica gel (70–230 mesh) using a mix of dichloromethane, light petroleum, and ethyl acetate as eluent in determined proportions. The resulting solution was concentrated to dryness under reduced pressure.

#### 5.2.5. Structural Characterization for Compounds (**3**–**47**)

The 6-Ethyl-2,4-dimethylpyrimido[4,5-*c*]isoquinoline-1,3,7,10(2*H*,4*H*)-tetraone(**3**): Prepared from 1-(2,5-dihydroxyphenyl)propan-1-one **1a** and **2**; yellow solid; mp 167.6–167.9 °C; ^1^H-NMR (CDCl_3_, 400 MHz) δ 7.11 (d, *J* = 10.3 Hz, 1H, H-9), 6.81 (d, *J* = 10.3 Hz, 1H, H-8), 3.76 (s, 3H, 2-*N*CH_3_), 3.47 (s, 3H, 4-*N*CH_3_), 3.40 (q, *J* = 7.3 Hz, 2H, 6-CH_2_CH_3_), 1.34 (t, *J* = 7.3 Hz, 3H, 6-CH_2_CH_3_); ^13^C-NMR (CDCl_3_,100 MHz) δ 185.0 (1C, C-10), 183.9 (1C, C-7), 171.2 (1C, C-6), 159.0 (1C, C-4a), 152.9 (1C, C-1), 151.5 (1C, C-3), 146.6 (1C, C-10a), 138.7 (1C, C-8), 138.7 (1C, C-9), 121.2 (1C, C-6a), 105.4 (1C, C-10b), 32.0 (1C, 6-CH_2_CH_3_), 30.6 (1C, 2-*N*CH_3_), 29.5 (1C, 4-*N*CH_3_), 12.5 (1C, 6-CH_2_CH_3_); HRMS *m*/*z* 299.09070 (Calculated for C_15_H_13_N_3_O_4_ [M]^+^, 299.09061); purified in column chromatography with dichloromethane: ethyl acetate = 9:1; yield: 84%.

The 2,4,6-Trimethylpyrimido[4,5-*c*]isoquinoline-1,3,7,10(2*H*,4*H*)-tetraone (**4**): Prepared from 1-(2,5-dihydroxyphenyl)ethan-1-one **1b** and **2**; yellow solid; mp 197.5–198.5 °C (d); ^1^H-NMR (CDCl_3_,400 MHz) δ 7.13 (d, *J* = 10.5 Hz, 1H, 8-H), 6.83 (d, *J* = 10.5 Hz, 1H, 9-H), 3.75 (s, 3H, 2-*N*CH_3_), 3.47 (s, 3H, 4-*N*CH_3_), 2.99 (s, 3H, 6-CH_3_); ^13^C-NMR (CDCl_3_,100 MHz) δ 184.2, 183.4, 166.2, 158.3, 152.3, 150.9, 145.8, 138.4, 138.1, 121.1, 105.2, 30.1, 28.9, 26.6; HRMS *m*/*z* 285.0828 (Calculated forC_14_H_11_N_3_O_4_ [M + H]^+^: 286.0832); purified by column chromatography, dichloromethane: ethyl acetate = 9:1; yield: 86%.

The 2,4-Dimethylpyrimido[4,5-*c*]isoquinoline-1,3,7,10(2*H*,4*H*)-tetraone (**5**): Prepared from 2,5-dihydroxybenzaldehyde **1c** and **2**; yellow solid; mp 203.5–205.5 °C (d);^1^H-NMR (400 MHz, CDCl_3_) δ 9.30 (s, 1H, 6-H), 7.15 (d, *J* = 10.5 Hz, 1H, 9-H), 6.88 (d, *J* = 10.5 Hz, 1H, 8-H), 3.79 (s, 3H, 2-*N*CH_3_), 3.51 (s, 3H, 4-*N*CH_3_); ^13^C-NMR (100 MHz, CDCl_3_) δ 182.8, 182.1, 158.2, 154.8, 153.1, 150.8, 142.9, 140.6, 136.3, 122.6, 106.3, 30.5, 29.1; HRMS *m*/*z* 272.0671 (Calculated for C_13_H_9_N_3_O_4_ [M + H]^+^: 272.06); purified by column chromatography with a mixture of dichloromethane: ethyl acetate = 9:1; yield: 86%.

The 8-(4-Aminobenzenethio)-6-ethyl-2,4-dimethylpyrimido[4,5-*c*]isoquinoline-1,3,7,10(2*H*,4*H*)-tetraone (**6**).A suspension of **11** (150.0 mg, 0.33 mmol), iron powder (370 mg, 6.63 mmol) in a 1:1:1 mixture of water/methanol/acetic acid (30 mL) was stirred for 1 h at 50–60 °C. The mixture was neutralized with NaHCO_3_ and then extracted with ethyl acetate (2 × 15 mL). The organic extract was dried over anhydrous Na_2_SO_4_, filtered, and evaporated under reduced pressure. The organic crude was purified using 45g of silica gel 60 (230–400 mesh). The resulting solution was concentrated to dryness under reduced pressure; brown solid; mp > 250 °C; ^1^H-NMR (400 MHz, CDCl_3_) δ 7.24 (d, *J* = 8.5 Hz, 2H, 3′-H y 5′-H), 6.73 (d, *J* = 8.5 Hz, 2H, 2′-H y 6′-H), 6.22 (s, 1H, 9-H), 4.01 (s, 2H, 4′-NH_2_), 3.74 (s, 3H, 2-*N*CH_3_), 3.42 (s, 3H, 4-*N*CH_3_), 3.40 (q, *J* = 7.3 Hz, 2H, 6-CH_2_CH_3_), 1.35 (t, *J* = 7.3 Hz, 3H, 6-CH_2_CH_3_); ^13^C-NMR (100 MHz, CDCl_3_): δ 181.5, 181.2, 170.7, 158.5, 158.1, 152.7, 151.1, 148.9, 147.5, 137.0 (2C), 127.7, 120.8, 116.3 (2C), 113.4, 105.4, 31.7, 30.2, 29.0, 12.2. HRMS *m*/*z* 423.1125 (Calculated for C_21_H_19_N_4_O_4_S[M + H]^+^: 423.1127); purified by column chromatography with a mixture of dichloromethane: ethyl acetate: petroleum ether = 9:1:1; yield: 31%.

The 8-(4-Acetamidobenzenethio)-6-ethyl-2,4-dimethylpyrimido[4,5-*c*]isoquinoline-1,3,7,10(2*H*,4*H*)-tetraone (**7**): Prepared from **3** and 4-acetamidothiophenol using general procedure B; orange solid; mp 170.2–173.0 °C; ^1^H-NMR (400 MHz, CDCl_3_) δ 7.99 (s, 1H, *N*HCO), 7.69 (d, *J* = 8.3 Hz, 2H, 3′-H and 5′-H), 7.43 (d, *J* = 8.4 Hz, 2H, 2′-H and 6′-H), 6.15 (s, 1H, 9-H), 3.75 (s, 3H, 2-*N*CH_3_), 3.42 (s, 3H, 4-*N*CH_3_), 3.40 (q, *J* = 7.5 Hz, 2H, 6-CH_2_CH_3_), 2.21 (s, 3H, COCH_3_), 1.36 (t, *J* = 7.3 Hz, 3H, 6-CH_2_CH_3_); ^13^C-NMR (100 MHz, CDCl_3_): δ 181.3, 180.8 (2C), 170.8, 168.9, 158.5, 157.2, 152.7, 151.0, 147.2, 140.6, 136.5 (2C), 127.7, 121.1 (2C), 120.6, 105.4, 31.7, 30.2, 29.1, 24.7, 12.1; HRMS *m*/*z* 465.1246 (Calculated for C_23_H_21_N_4_O_5_S [M + H]^+^: 465.1233); purified in column chromatography with dichloromethane: ethyl acetate = 3:1; yield: 69%.

The 8-(4-hydroxybenzenethio)-6-ethyl-2,4-dimethylpyrimido[4,5-*c*]isoquinoline-1,3,7,10(2*H*,4*H*)-tetraone(**8**): Prepared from **3** and 4-mercaptophenol using general procedure B; orange solid; mp 208–210 °C (d); ^1^H-NMR (400 MHz, CDCl_3_) δ 7.38 (d, *J* = 8.7 Hz, 2H, 3′-H and 5′-H), 6.96 (d, *J* = 8.7 Hz, 2H,2′-H and 6′-H), 6.21 (s, 1H, 4′-OH), 6.15 (s, 1H, 9-H), 3.77 (s, 3H, 2-*N*CH_3_), 3.46 (s, 3H, 4-*N*CH_3_), 3.43 (q, *J* = 7.3 Hz, 2H, 6-CH_2_CH_3_), 1.38 (t, *J* = 7.3 Hz, 3H, 6-CH_2_CH_3_); ^13^C-NMR (100 MHz, CDCl_3_) δ 181.5, 181.0, 171.0, 159.1, 158.2, 157.6, 155.8, 152.7, 151.0, 147.4, 137.4 (2C), 127.7, 120.8, 117.6 (2C), 117.0, 31.8, 30.3, 29.2, 12.2; HRMS *m*/*z* 424.0963 (Calculated for C_21_H_18_N_3_O_5_S [M + H]^+^: 424.0967); purified in column chromatography with ethyl acetate: petroleum ether = 9:0.8; yield: 72%.

The 4-((6-ethyl-2,4-dimethyl-1,3,7,10-tetraoxo-1,2,3,4,7,10-hexahydropyrimido[4,5-*c*]isoquinolin-8-yl)thio)benzonitrile (**9**): Prepared from **3** and 4-mercaptobenzonitrile using general procedure B; orange solid; mp 203–205 °C; ^1^H-NMR (400 MHz, CDCl_3_) δ 7.80 (d, *J* = 8.0 Hz, 2H, 3′-H and 5′-H), 7.69 (d, *J* = 8.0 Hz, 2H, 2′-H and 6′-H), 6.21 (s, 1H, 9-H), 3.76 (s, 3H, 2-*N*CH_3_), 3.44 (s, 3H, 4-*N*CH_3_), 3.41 (q, *J* = 7.3 Hz, 2H, 6-CH_2_CH_3_), 1.37 (t, *J* = 7.3 Hz, 3H, 6-CH_2_CH_3_); ^13^C-NMR (100 MHz, CDCl_3_) δ 181.1, 180.2, 171.0, 158.3, 158.2, 154.6, 152.9, 151.0, 146.9, 136.2 (2C), 134.0, 133.7 (2C) 128.4, 120.3, 117.6, 114.6, 105.5, 31.8, 30.2, 29.1, 12.1; HRMS *m*/*z* 433.0892 (Calculated for C_22_H_17_N4O_4_S [M + H]^+^: 433.0971); purified in column chromatography with dichloromethane: ethyl acetate = 9:1; yield: 72%.

Synthesis of 8-(4-carboxibenzenethio)-6-ethyl-2,4-dimethylpyrimido[4,5-*c*]isoquinoline-1,3,7,10(2*H*,4*H*)-tetraone (**10**). Prepared from **3** and 4-mercaptobenzoic acid; A solution of **3** (150 mg, 0.4909 mmol) and CeCl_3_∙7H_2_O (5% mmol respect to **3**) in a mix of ethanol: dichloromethane = 1:1 (10 mL), was added dropwise slowly a solution of 4-mercaptobenzoic acid (0.5 equiv.) in ethanol: dichloromethane = 1:1 (30 mL). The reaction mixture was stirred at room temperature for 16 h. The progress of the reaction was followed by thin-layer chromatography (TLC). Then, 10 mL of distilled water and NaOH (0.1 M) are added to the solution until pH 10 is reached. The extractions were carried out with ethyl acetate (10 mL × 2), the precipitate solid, filtered under vacuum, and washed with ethanol (15 mL × 3). Finally, the obtained product was recrystallized from ethanol. Orange solid, mp > 250 °C; ^1^H-NMR (400 MHz DMSO_6_) δ 13.29 (s, 1H, 4′-COOH), 8.10 (d, 2H, 3′-H and 5′-H), 7.76 (d, 2H, 2′-H and 6′-H), 6.07 (s, 1H, 9-H), 3.59 (s, 3H, 2-*N*CH_3_), 3.30 (q, *J* = 7.3 Hz, 2H, 6-CH_2_CH_3_) 3.23 (s, 3H, 4-*N*CH_3_), 1.29 (t, *J* = 7.3 Hz, 3H, 6-CH_2_CH_3_); ^13^C-NMR (100 MHz DMSO_6_) δ 181.1, 180.4, 169.3, 167.0, 158.3, 155.1, 152.9, 151.2, 146.4, 135.8, 133.1, 133.0, 131.5 (2C), 127.8 (2C), 120.6, 105.8, 31.4, 30.3, 29.0, 12.3; HRMS *m*/*z* 452.0912 (Calculated for C_22_H_18_N_3_O_6_S [M + H]^+^: 452.0916); yield: 38%.

The 8-(4-nitrobenzenethio)-6-ethyl-2,4-dimethyl-pyrimido[4,5-*c*]isoquinoline-1,3,7,10(2*H*,4*H*)-tetraone (**11**): Prepared from **3** and 4-nitrobenzenethiol using general procedure B; yellow solid; mp 188.8–190.1 °C (d); ^1^H-NMR (400 MHz, CDCl_3_) δ 8.34 (d, 2H, 3′-H and 5′-H), 7.75 (d, 2H, 2′-H and 6′-H), 6.24 (s, 1H, 9-H), 3.75 (s, 3H, 2-*N*CH_3_), 3.42 (s, 3H, 4-*N*CH_3_), 3.40 (q, *J* = 7.3 Hz, 2H, 6-CH_2_CH_3_), 1.36 (t, *J* = 7.3 Hz, 3H, 6-CH_2_CH_3_); ^13^C-NMR (100 MHz, CDCl_3_) δ 181.0, 181.0, 170.9, 158.3, 154.3, 152.9, 151.0, 149.1, 146.9, 136.4 (2C), 136.1, 128.6, 125.1 (2C), 120.2, 105.4, 31.8, 30.3, 29.1, 12.1; HRMS *m*/*z* 453.0871 (Calculated for C_21_H_17_N_4_O_6_S [M + H]^+^: 453.0869); purified in column chromatography with dichloromethane: ethyl acetate: petroleum ether = 10:1:4; yield: 96%.

The 8-(4-dimethylaminobenzenethio)-6-ethyl-2,4-dimethylpyrimido[4,5-*c*]isoquinoline-1,3,7,10(2*H*,4*H*)-tetraone (**12**): Prepared from **3** and 4-(dimethylamino)benzenethiol using general procedure B; burgundy red; mp 134.6–137.0 °C; ^1^H-NMR (400 MHz, CDCl_3_) δ 7.31 (d, *J* = 8.6 Hz, 2H, 2′-H and 6′-H), 6.75 (d,*J* = 8.6 Hz, 2H, 3′-H and 5′-H), 6.21 (s, 1H, 9-H), 3.74 (s, 3H, 2-*N*CH_3_), 3.43 (s, 3H, 4-*N*CH_3_), 3.40 (q, *J* = 7.3 Hz, 2H, 6-CH_2_CH_3_), 3.03 (s, 6H, 4′-*N*(CH_3_)_2_), 1.36 (t, *J* = 7.3 Hz, 3H, 6-CH_2_CH_3_);^13^C-NMR (100 MHz, CDCl_3_) δ 181.7, 181.5, 170.7, 158.7, 158.6, 152.8, 151.9, 151.3, 147.7, 136.7 (2C), 127.8, 121.0, 113.6 (2C), 110.6, 105.5, 77.2, 40.3, 31.8, 30.3, 29.2, 12.3; HRMS *m*/*z* 451.1447 (Calculated for C_23_H_23_N_4_O_4_S [M + H]^+^: 451.1440); purified in column chromatography with dichloromethane: ethyl acetate: petroleum ether = 9:0.5:1; yield: 43%.

The 8-(4-(methylthio)benzenethio)-6-ethyl-2,4-dimethylpyrimido[4,5-*c*]isoquinoline-1,3,7,10(2*H*,4*H*)-tetraone (**13**): Prepared from **3** and 4-(methylthio)benzenethiol using general procedure B; red solid; mp 181.6–184.0 °C; ^1^H-NMR (400 MHz, CDCl_3_) δ 7.41 (d, *J* = 8.1 Hz, 2H, 2′-H and 6′-H), 7.33 (d, *J* = 8.2 Hz, 2H, 2′-H and 6′-H), 6.19 (s, 1H, 9-H), 3.75 (s, 3H, 2-*N*CH_3_), 3.43 (s, 3H, 4-*N*CH_3_), 3.41 (q, *J* = 7.4 Hz, 2H, 6-CH_2_CH_3_), 2.53 (s, 3H, SCH_3_), 1.36 (t, *J* = 7.3 Hz, 3H, 6-CH_2_CH_3_); ^13^C-NMR (100 MHz, CDCl_3_) δ 181.33, 180.8, 170.7, 158.4, 156.8, 152.7, 151.1, 147.3, 143.1, 135.8 (2C), 127.9, 127.4 (2C), 122.4, 120.6, 105.4, 31.7, 30.2, 29.1, 15.2, 12.1; HRMS *m*/*z* 454.0894 (Calculated for C_22_H_20_N_3_O_4_S_2_ [M + H]^+^: 454.0895); purified in column chromatography with dichloromethane: ethyl acetate: petroleum ether = 3:2:0.5; yield: 46%.

The 6-ethyl-8-(4-ethylbenzenethio)-2,4-dimethylpyrimido[4,5-*c*]isoquinoline-1,3,7,10(2*H*,4*H*)-tetraone (**14**): Prepared from **3** and 4-ethylbenzenethiol using general procedure B; yellow solid; mp 172–174 °C; ^1^H-NMR (400 MHz, CDCl_3_) δ 7.43 (d, *J* = 7.9 Hz, 2H, 3′-H and 5′-H), 7.33 (d, *J* = 7.9 Hz, 2H, 2′-H and 6′-H), 6.18 (s, 1H, 9-H), 3.75 (s, 3H, 2-*N*CH_3_), 3.43 (s, 3H, 4-*N*CH_3_), 3.41 (q, *J* = 7.3 Hz, 2H, 6-CH_2_CH_3_), 2.72 (q, *J* = 7.6 Hz, 2H, 4′-CH_2_CH_3_), 1.37 (t, *J* = 7.3 Hz, 3H, 4′-CH_2_CH_3_), 1.28 (t, *J* =7.6 Hz, 3H, 6-CH_2_CH_3_); ^13^C-NMR (100 MHz, CDCl_3_) δ 181.4, 180.9, 170.7, 158.4, 157.2, 152.7, 151.1, 147.4, 135.7 (2C), 130.1 (2C), 127.8, 123.7, 120.6, 105.4, 31.7, 30.2, 29.0, 28.7, 15.3, 12.2; HRMS *m*/*z* 436.1253 (Calculated for C_23_H_22_N_3_O_4_S [M + H]^+^: 436.1331); purified in column chromatography with dichloromethane: ethyl acetate: petroleum ether = 12:0.5:5; yield: 83%.

The 6-ethyl-2,4-dimethyl-8-((4-trifluoromethyl)phenyl)thio)pyrimido[4,5-*c*]isoquinoline-1,3,7,10(2*H*,4*H*)-tetraone(**15**): Prepared from **3** and 4-(trifluoromethyl)benzenethiol using general procedure B; yellow solid; mp 204–206 °C; ^1^H-NMR (400 MHz, CDCl_3_) δ 7.77 (d, *J* = 8.1 Hz, 2H, 3′-H and 5′-H), 6.96 (d, *J* = 8.1 Hz, 2H, 2′-H and 6′-H), 6.18 (s, 1H, 9-H), 3.76 (s, 3H, 2-*N*CH_3_), 3.43 (s, 3H, 4-*N*CH_3_), 3.42 (q, *J* = 7.3 Hz, 2H, 6-CH_2_CH_3_), 1.37 (t, *J* = 7.3 Hz, 3H, 6-CH_2_CH_3_); ^13^C-NMR (100 MHz, CDCl_3_) δ 181.2, 181.4, 170.9, 158.3, 155.3, 152.8, 151.0, 147.1, 136.1 (2C), 132.8 (q, *J* = 33.1 Hz, 1C), 132.2, 128.2, 127.2 (q, *J* = 3.6 Hz, 2C), 123.5 (q, *J* = 272.8 Hz, 1C), 120.4, 105.5, 31.8, 30.2, 29.1, 12.1; HRMS *m*/*z* 476.0814 (Calculated for C_22_H_17_F_3_N_3_O_4_S [M + H]^+^: 476.0812); purified in column chromatography with dichloromethane: ethyl acetate: petroleum ether = 12:0.5:5; yield: 89%.

The 8-(4-(isopropyl)benzenethio)-6-ethyl-2,4-dimethylpyrimido[4,5-*c*]isoquinoline-1,3,7,10(2*H*,4*H*)-tetraone (**16**): Prepared from **3** and 4-isopropylbenzenethiol using general procedure B; orange solid; mp 134.6–137.0 °C; ^1^H-NMR (400 MHz, CDCl_3_) δ 7.43 (d, *J* = 8.0 Hz, 2H, 3′-H and 5′-H), 7.35 (d, *J* = 8.0 Hz, 2H, 2′-H and 6′-H), 6.17 (s, 1H, 9-H), 3.74 (s, 3H, 2-*N*CH_3_), 3.42 (s, 3H, 4-*N*CH_3_), 3.40 (q, *J* = 7.3 Hz, 2H, 6-CH_2_CH_3_), 3.03–2.90 (m, 1H, 4′-CH(CH_3_)_2_), 1.36 (t, *J* = 7.3 Hz, 3H, 6-CH_2_CH_3_), 1.28 (d, *J* = 6.9 Hz, 6H, 4′-CH(CH_3_)_2_); ^13^C-NMR (100 MHz, CDCl_3_) δ 181.6, 181.0, 170.8, 158.5, 157.3, 152.8, 152.1, 151.2, 147.5, 135.8(2C), 128.7 (2C), 127.9, 123.9, 120.7, 105.5, 34.2, 31.8, 30.2, 29.1, 23.9 (2C), 12.2; HRMS *m*/*z* 450.1499 (Calculated for C_24_H_24_N_3_O_4_S [M + H]^+^: 450.1488); purified in column chromatography with dichloromethane: ethyl acetate: petroleum ether = 12:0.5:1; yield: 43%.

The 8-(4-(tertbutyl)benzenethio)-6-ethyl-2,4-dimethylpyrimido[4,5-*c*]isoquinoline-1,3,7,10(2*H*,4*H*)-tetraone (**17**): Prepared from **3** and 4-(*tert*-butyl)benzenethiol using general procedure B; orange solid, mp 157.6–160.7 °C; ^1^H-NMR (400 MHz, CDCl_3_) δ 7.52 (d, *J* = 8.2 Hz, 2H, 3′-H and 5′-H), 7.45 (d, *J* = 8.1 Hz, 2H, 2′-H and 6′-H), 6.19 (s, 1H, 9-H), 3.75 (s, 3H, 2-*N*CH_3_), 3.43 (s, 3H, 4-*N*CH_3_), 3.42 (q, *J* = 7.3 Hz, 2H, 6-CH_2_CH_3_), 1.37 (t, *J* = 7.4 Hz, 3H, 6-CH_2_CH_3_), 1.36 (s, 9H, 4′-C(CH_3_)_3_); ^13^C-NMR (100 MHz, CDCl_3_) δ 181.7, 181.0, 170.9, 158.6, 157.3, 154.4, 152.8, 151.2, 147.5, 135.5(2C), 128.0, 127.9, 127.7, 126.2, 123.7, 120.7, 105.5, 35.1, 31.8, 31.4, 31.3, 30.3, 29.2, 12.3; HRMS *m*/*z* 464.1653 (Calculated for C_25_H_26_N_3_O_4_S [M + H]^+^: 464.1644); purified in column chromatography with dichloromethane: ethyl acetate: petroleum ether = 12:0.5:5; yield: 61%.

The 8-(benzylthio)-6-ethyl-2,4-dimethylpyrimido[4,5-*c*]isoquinoline-1,3,7,10(2*H*,4*H*)-tetraone (**18**): Prepared from **3** and phenylmethanethiol using general procedure B; orange solid; mp 181.0–182.0 °C; ^1^H-NMR (400 MHz, CDCl_3_) δ 7.40–7.28 (m, 5H, C_6_H_5_), 6.76 (s, 1H, 9-H), 4.06 (s, 2H, Ph-CH_2_-S), 3.74 (s, 3H, 2-*N*CH_3_), 3.44 (s, 3H, 4-*N*CH_3_), 3.36 (q, *J* = 7.3 Hz, 2H, 6-CH_2_CH_3_), 1.33 (t, *J* = 7.3 Hz, 3H, 6-CH_2_CH_3_); ^13^C-NMR (100 MHz, CDCl_3_) δ 180.8 (2C), 170.8, 158.5, 154.6, 152.8, 151.2, 147.1, 134.0, 129.1 (2C), 129.0 (2C), 128.2, 127.1, 120.7, 105.4, 35.8, 31.8, 30.2, 29.1, 12.2. HRMS *m*/*z* 422.1171 (Calculated for C_22_H_20_N_3_O_4_S [M + H]^+^: 422.1175); purified in column chromatography with dichloromethane: ethyl acetate: petroleum ether = 9:1:3; yield: 66%.

The 8-(phenethylthio)-6-ethyl-2,4-dimethyl-pyrimido[4,5-*c*]isoquinoline-1,3,7,10(2*H*,4*H*)-tetraone (**19**): Prepared from **3** and 2-phenylethanethiol using general procedure B; yellow solid; mp 170.0–171.0 °C; ^1^H-NMR (400 MHz, CDCl_3_) δ 7.33 (t, *J* = 7.3 Hz, 2H, 3′-H and 5′-H), 7.26–7.24 (m, 3H, 2′-H, 4′-H and 6′-H), 6.70 (s, 1H, 9-H), 3.75 (s, 3H, 2-*N*CH_3_), 3.46 (s, 3H, 4-*N*CH_3_), 3.37 (q, *J* = 7.3 Hz, 2H, 6-CH_2_CH_3_), 3.13–3.00 (m, 4H, Ph-C_2_H_4_S), 1.34 (t, *J* = 7.3 Hz, 3H, 6-CH_2_CH_3_); ^13^C-NMR (100 MHz, CDCl_3_) δ 180.7 (2C), 170.7, 158.5, 154.9, 152.7, 151.1, 147.0, 138.9, 128.8 (2C), 128.5 (2C), 127.0, 126.5, 120.7, 105.4, 33.7, 32.3, 31.7, 30.2, 29.1, 12.1; HRMS *m*/*z* 436.1332 (Calculated for C_23_H_22_N_3_O_4_S [M + H]^+^: 436.1331); purified in column chromatography with dichloromethane: ethyl acetate: petroleum ether = 10:1:6; yield: 79%.

The (4-(chlorobenzyl)thio)-6-ethyl-2,4-dimethylpyrimido[4,5-*c*]isoquinoline-1,3,7,10(2*H*,4*H*)-tetraone (**20**): Prepared from **3** and (4-chlorophenyl)methanethiol using general procedure B; orange solid; mp 191.0–191.8 °C; ^1^H-NMR (400 MHz, CDCl_3_) δ 7.33 (m, 4H, 2′-H, 3′-H, 5′-H, and 6′-H), 6.72 (s, 1H, 9-H), 4.02 (s, 2H, Ph-CH_2_-S), 3.74 (s, 3H, 2-*N*CH_3_), 3.45 (s, 3H, 4-*N*CH_3_), 3.36 (q, *J* = 7.3 Hz, 2H, 6-CH_2_CH_3_), 1.33 (t, *J* = 7.3 Hz, 3H, 6-CH_2_CH_3_); ^13^C-NMR (100 MHz, CDCl_3_) δ 180.7, 180.6, 170.7, 158.4, 154.1, 152.7, 151.0, 146.9, 134.0, 132.4, 130.2 (2C), 129.2 (2C), 127.1, 120.6, 105.4, 35.0, 31.7, 30.2, 29.1, 12.1; HRMS *m*/*z* 456.0775 (Calculated for C_22_H_19_ClN_3_O_4_S [M + H]^+^: 456.0785); purified in column chromatography with dichloromethane: ethyl acetate: petroleum ether = 10:1:5; yield: 32%.

The 8-(benzo[*d*]thiazol-2-ylthio)-6-ethyl-2,4-dimethylpyrimido[4,5-*c*]isoquinoline-1,3,7,10(2*H*,4*H*)-tetraone(**21**): Prepared from **3** and benzo[*d*]thiazole-2-thiol using general procedure B; yellow solid; mp > 250 °C; ^1^H-NMR (400 MHz, CDCl_3_) δ 8.07 (d, *J* = 8.1 Hz, 1H, 7′-H), 7.89 (d, *J* = 8.1 Hz, 1H, 4′-H), 7.54 (d, *J* = 7.6 Hz, 1H, 5′-H), 7.47 (d, *J* = 7.6 Hz, 1H, 6′-H), 6.25 (s, 1H, 9-H), 3.74 (s, 3H, 2-*N*CH_3_), 3.42 (s, 3H, 4-*N*CH_3_), 3.38 (q, *J* = 7.3 Hz, 2H, 6-CH_2_CH_3_), 1.34 (t, *J* = 7.3 Hz, 3H, 6-CH_2_CH_3_); ^13^C-NMR (100 MHz, CDCl_3_) δ 181.5, 179.9, 170.9, 158.2, 156.7, 153.6, 152.9, 151.0, 150.5, 146.8, 136.9, 131.4, 126.7, 123.6, 121.5, 105.4 31.7, 30.2, 29.1, 12.1; HRMS *m*/*z* 465.0690 (Calculated for C_22_H_17_N_4_O_4_S _2_ [M + H]^+^: 465.0691); purified in column chromatography with dichloromethane: ethyl acetate: petroleum ether = 9:1:3; yield: 72%.

The 8-(phenylamino)-6-ethyl-2,4-dimethyl-pyrimido[4,5-*c*]isoquinoline-1,3,7,10(2*H*,4*H*)-tetraone (**22**): Prepared from **3** and aniline using general procedure C; purple solid; mp 189.0–190.0 °C; ^1^H -NMR (400 MHz, CDCl_3_) δ 7.60 (s, 1H, *N*H), 7.42 (t, *J* = 7.8 Hz, 2H, 3′-H and 5′-H), 7.23 (m, 3H, 2′-H, 4′-H and 6′-H), 6.46 (s, 1H, 9-H), 3.76 (s, 3H, 2-*N*CH_3_), 3.47 (s, 3H, 4-*N*CH_3_), 3.41 (q, *J* = 7.3 Hz, 2H, 6-CH_2_CH_3_), 1.37 (t, *J* = 7.3 Hz, 3H, 6-CH_2_CH_3_); ^13^C-NMR (100 MHz, CDCl_3_) δ 182.2, 180.0, 170.2, 158.7, 153.1, 151.2, 149.3, 144.6, 137.2, 129.8(2C), 125.8, 122.3(2C), 119.5, 105.9, 103.7, 31.8, 30.2, 29.1, 12.1; HRMS *m*/*z* 391.1412 (Calculated for C_21_H_19_N_4_O_4_ [M + H]^+^: 391.1406); purified in column chromatography with dichloromethane: ethyl acetate: petroleum ether = 1:2:4; yield: 76%.

Synthesis of 8-(4-amino-phenylamino)-6-ethyl-2,4-dimethylpyrimido[4,5-*c*]isoquinoline-1,3,7,10(2*H*,4*H*)-tetraone (**23**): A solution of **3** (150 mg, 0.4909 mmol) and CeCl_3_^*^7H_2_O (5% mmol respect to **1**) in a mix of ethanol: dichloromethane = 1:1 (10 mL), was added dropwise slowly a solution of benzene-1,4-phenylendiamine (26.60 mg, 0.2454 mmol) in ethanol: dichloromethane = 1:1 (30mL). The reaction mixture was stirred at room temperature for 16 h. The progress of the reaction was followed by thin-layer chromatography (TLC). The reaction mixture was concentrated under reduced pressure, and the crude reaction was purified using 30 g of silica gel (70–230 mesh) and a mix of chloroform and ethyl acetate as eluent; green solid; mp > 250 °C; ^1^H-NMR (400 MHz, CDCl_3_) δ 7.26 (d, *J* = 8.3 Hz, 2H, 2′-H and 6′-H), 7.26 (s, 1H, NH), 6.75 (d, *J* = 8.4 Hz, 2H, 3′-H and 5′-H), 6.23 (s, 1H, 9-H), 3.99 (s, 2H, 4′-NH_2_), 3.75 (s, 3H, 2-*N*CH_3_), 3.44 (s, 3H, 4-*N*CH_3_), 3.41 (q, *J* = 7.3 Hz, 2H, 6-CH_2_CH_3_), 1.36 (t, *J* = 7.3 Hz, 3H, 6-CH_2_CH_3_); ^13^C-NMR (100 MHz, CDCl_3_) δ 181.5, 181.2, 170.7, 158.5, 158.1, 152.7, 151.1, 148.9, 147.5, 137.0 (2C), 127.7, 120.8, 116.4 (2C), 113.5, 105.4, 31.7, 30.2, 29.0, 12.2; HRMS *m*/*z* 406.1528 (Calculated for C_21_H_20_N_5_O_4_ [M + H]^+^: 406.1515); purified in column chromatography with chloroform: ethyl acetate = 8:1; yield: 51%.

The 8-(4-(methoxycarbonyl)phenylamino)-6-ethyl-2,4-dimethylpyrimido[4,5-c]isoquinoline-1,3,7,10(2*H*,4*H*)-tetraone (**24**): Prepared from **3** and methyl 4-aminobenzoate using general procedure C; red solid; mp > 250 °C; ^1^H-NMR (400 MHz, CDCl_3_) δ 8.10 (d, *J* = 8.1 Hz, 2H, 3′-H and 5′-H), 7.76 (s, 1H, *N*H), 7.31 (d, *J* = 8.1 Hz, 2H, 2′-H and 6′-H), 6.63 (s, 1H, 9-H), 3.93 (s, 3H, 4′-COOCH_3_), 3.77 (s, 3H, 2-*N*CH_3_), 3.47 (s, 3H, 4-*N*CH_3_), 3.41 (q, *J* = 7.3 Hz, 2H, 6-CH_2_CH_3_), 1.38 (t, *J* = 7.3 Hz, 3H, 6-CH_2_CH_3_); ^13^C-NMR (100 MHz, CDCl_3_) δ 182.4, 179.6, 170.4, 155.4, 151.9, 143.3, 141.6, 134.5, 132.6 (2C), 131.4, 123.9 (2C), 120.6, 118.9, 110.4, 108.9, 105.5, 57.6, 31.9, 30.2, 29.1, 12.0; HRMS *m*/*z* 449.1469 (Calculated for C_23_H_21_N_4_O_6_ [M + H]^+^: 449.1461); purified in column chromatography with chloroform: ethyl acetate = 20:1; yield: 42%.

Synthesis of methyl 4-(6-ethyl-2,4-dimethyl-1,3,7,10-tetraoxo-1,2,3,4,7,10-hexahydropyrimido[4,5-*c*]isoquinolin-8-yl)amino)benzoic acid (**25**). Prepared from **3** and 4-aminobenzoic acid; A solution of **3** (150 mg, 0.4909 mmol) and CeCl_3_^*^7H_2_O (5% mmol respect to **3**) in a mix of ethanol: dichloromethane = 1:1 (10 mL), was added dropwise slowly a solution of 4-aminobenzoic acid (34.40 mg, 0.2506 mmol) in ethanol: dichloromethane = 1:1 (30 mL). The reaction mixture was stirred at room temperature for 16 h. The progress of the reaction was followed by thin-layer chromatography (TLC). The reaction mixture was concentrated under reduced pressure, and the obtained solid was washed three times with 30 mL of dichloromethane. Finally, the solid was purified using 10 g of silica gel (70–230 mesh) and ethyl acetate as eluent; red solid; mp > 250 °C; ^1^H-NMR (400 MHz DMSO-d_6_) δ 9.47 (s, 1H, *N*H), 7.97 (d, *J* = 8.6 Hz, 2H, 3′-H and 5′-H), 7.54 (d, *J* = 8.7 Hz, 2H, 2′-H and 6′-H), 6.40 (s, 1H, 9-H), 3.61 (s, 3H, 2-*N*CH_3_), 3.27 (q, *J* = 7.5 Hz, 2H, 6-CH_2_CH_3_) 3.23 (s, 3H, 4-*N*CH_3_), 1.32 (t, *J* = 7.3 Hz, 3H, 6-CH_2_CH_3_); ^13^C-NMR (400 MHz DMSO_6_): Not obtained due to low solubility of the compound; HRMS *m*/*z* 435.1299 (Calculated for C_22_H_19_N_4_O_6_ [M + H]^+^: 435.1305); purified in column chromatography with ethyl acetate; yield: 12%.

The 6-ethyl-8-((4-fluorophenyl)amino)-2,4-dimethylpyrimido[4,5-*c*]isoquinoline-1,3,7,10(2*H*,4*H*)-tetraone (**26**): Prepared from **3** and 4-fluoroaniline using general procedure C; burgundy red solid; mp 216.1–216.9 °C (d); ^1^H-NMR (400 MHz, CDCl_3_) δ7.49 (s, 1H, *N*H), 7.22 (dt, *J*_H,H_ = 7.9, *J*_F,H_ = 2.6 Hz, 2H, 2′-H and 6′-H), 7.13 (t, *J*_H,H_ = 8.5, *J*_F,H_ = 8.5 Hz, 2H, 3′-H and 5′-H), 6.30 (s, 1H, 9-H), 3.76 (s, 3H, 2-*N*CH_3_), 3.47 (s, 3H, 4-*N*CH_3_), 3.41 (q, *J* = 7.3 Hz, 2H, 6-CH_2_CH_3_), 1.37 (t, *J* = 7.3 Hz, 3H, 6-CH_2_CH_3_); ^13^C-NMR (100 MHz, CDCl_3_) δ 182.1, 179.9, 170.2, 160.4 (d, 1C, *J*_F,C_ = 253.5 Hz, 4′), 158.8, 153.1, 151.2, 149.3, 145.1, 133.1(d, 1C, *J*_F,C_ = 2.9 Hz, 1′), 124.7 (d, 2C, *J*_F,C_ = 8.3 Hz, 2′ and 6′), 119.4, 116.8 (d, 2C, *J*_F,C_ = 22.8 Hz, 3′ and 5′), 106.0, 103.3, 31.8, 30.2, 29.1, 12.1; HRMS *m*/*z* 409.1307 (Calculated for C_21_H_18_FN_4_O_4_ [M + H]^+^: 409.1312); purified in column chromatography with chloroform: ethyl acetate = 9:1; yield: 70%.

The 8-((4-chlorophenyl)amino)-6-ethyl-2,4-dimethylpyrimido[4,5-*c*]isoquinoline-1,3,7,10(2*H*,4*H*)-tetraone (**27**); Prepared from **3** and 4-chloroaniline using general procedure C; purple solid; mp 206.0–207.0 °C (d); ^1^H-NMR (400 MHz, CDCl_3_) δ 7.56 (s,1H, *N*H), 7.39 (d, *J* = 8.8 Hz, 2H, 3′-H and 5′-H), 7.20 (d, *J* = 8.8 Hz, 2H, 2′-H and 6′-H), 6.40 (s, 1H, 9-H), 3.76 (s, 3H, 2-*N*CH_3_), 3.47 (s, 3H, 4-*N*CH_3_), 3.40 (q, *J* = 7.3 Hz, 2H, 6-CH_2_CH_3_), 1.36 (t, *J* = 7.3 Hz, 3H, 6-CH_2_CH_3_); ^13^C-NMR (100 MHz, CDCl_3_) δ 182.2, 179.8, 170.3, 158.6, 153.1, 151.2, 149.1, 144.3, 135.8, 131.0, 129.3 (2C), 123.5 (2C), 119.3, 105.9, 104.0, 31.8, 30.2, 29.1, 12.1; HRMS *m*/*z* 425.1021 (Calculated for C_21_H_18_ClN_4_O_4_ [M + H]^+^: 425.1017); purified in column chromatography with chloroform: ethyl acetate: petroleum ether = 2:1:2; yield: 53%.

The 8-((4-bromophenyl)amino)-6-ethyl-2,4-dimethylpyrimido[4,5-*c*]isoquinoline-1,3,7,10(2*H*,4*H*)-tetraone (**28**): Prepared from **3** and 4-bromoaniline using general procedure C; red solid; mp > 250.0 °C; ^1^H-NMR (400 MHz, CDCl_3_) δ7.55 (s, 1H, *N*H), 7.54 (d, *J* = 8.7 Hz, 2H, 3′-H and 5′-H), 7.15 (d, *J* = 8.7 Hz, 2H, 2′-H and 6′-H), 6.42 (s, 1H, 9-H), 3.76 (s, 3H, 2-*N*CH_3_), 3.47 (s, 3H, 4-*N*CH_3_), 3.40 (q, *J* = 7.3 Hz, 2H, 6-CH_2_CH_3_), 1.36 (t, *J* = 7.3 Hz, 3H, 6-CH_2_CH_3_);^13^C-NMR (100 MHz, CDCl_3_) δ 182.2, 179.8, 170.3, 158.6, 153.1, 151.2, 149.1, 144.2, 136.4, 132.9 (2C), 123.7 (2C), 119.3, 118.7, 105.9, 104.1, 31.8, 30.2, 29.1, 12.1; HRMS *m*/*z* 469.0515 (Calculated for C_21_H_18_BrN_4_O_4_ [M + H]^+^: 469.0511); purified in column chromatography with dichloromethane: ethyl acetate: petroleum ether = 4:1:4; yield: 67%.

The 6-ethyl-8-((4-iodophenyl)amino)-2,4-dimethylpyrimido[4,5-*c*]isoquinoline-1,3,7,10(2*H*,4*H*)-tetraone (**29**): Prepared from **3** and 4-iodoaniline using general procedure C; purple solid; mp > 250 °C; ^1^H-NMR (400 MHz, CDCl_3_) δ 7.73 (d, *J* = 8.1 Hz, 3H, 3′-H and 5′-H), 7,55 (s, 1H, *N*H), 7.02 (d, *J* = 8.2 Hz, 2H, 2′-H and 6′-H), 6.43 (s, 1H, 9-H), 3.76 (s, 3H, 2-*N*CH_3_), 3.47 (s, 3H, 4-*N*CH_3_), 3.40 (q, *J* = 7.3 Hz, 2H, 6-CH_2_CH_3_), 1.36 (t, *J* = 7.3 Hz, 3H, 6-CH_2_CH_3_);^13^C-NMR (100 MHz, CDCl_3_) δ 182.2, 179.8, 170.3, 158.6, 153.1, 151.2, 149.4, 144.0, 138.8 (2C), 137.1, 123.8 (2C), 119.3, 105.9, 104.3, 89.3, 31.8, 30.2, 29.1, 12.0; HRMS *m*/*z* 517.0372 (Calculated for C_21_H_18_IN_4_O_4_ [M + H]^+^: 517.0373); purified in column chromatography with chloroform; yield: 94%.

The 6-ethyl-8-(ethylthio)-2,4-dimethylpyrimido[4,5-*c*]isoquinoline-1,3,7,10(2*H*,4*H*)-tetraone (**30**); Prepared from **3** and ethanethiol; orange solid; mp 171.3–172.8 °C; ^1^H-NMR (400 MHz, CDCl_3_) δ 6.67 (s, 1H, 9-H), 3.75 (s, 3H, 2-*N*CH_3_), 3.46 (s, 3H, 4-*N*CH_3_), 3.37 (q, *J* = 7.3 Hz, 2H, 6-CH_2_CH_3_), 2.85 (q, *J* = 7.4 Hz, 2H, 8-S-CH_2_CH_3_), 1.43 (t, *J* = 7.4 Hz, 3H, 8-S-CH_2_CH_3_), 1.33 (t, *J* = 7.3 Hz, 3H, 6-CH_2_CH_3_); ^13^C-NMR (100 MHz, CDCl_3_) δ 180.8, 180.7, 170.7, 158.5, 155.1, 152.7, 151.1, 147.1, 126.5, 120.8, 105.4, 31.7, 30.2, 29.1, 24.9, 12.5 12.1; HRMS *m*/*z* 360.1010 (Calculated for C_17_H_18_N_3_O_4_S [M + H]^+^: 360.1018); purified in column chromatography with dichloromethane: ethyl acetate: petroleum ether = 15:3:4; yield: 48%.

The 6-ethyl-2,4-dimethyl-8-(propylthio)pyrimido[4,5-*c*]isoquinoline-1,3,7,10(2*H*,4*H*)-tetraone (**31**): Prepared from **3** and propane-1-thiol using general procedure B; orange solid; mp 163.8–164.9 °C; ^1^H-NMR (400 MHz, CDCl_3_) δ 6.68 (s, 1H, 9-H), 3.76 (s, 3H, 2-*N*CH_3_), 3.48 (s, 3H, 4-*N*CH_3_), 3.39 (q, *J* = 7.3 Hz, 2H, 6-CH_2_CH_3_), 2.81 (t, *J* =7.3 Hz, 2H, 8-S-CH_2_CH_2_CH_3_), 1.82 (m, 2H, 8-S-CH_2_CH_2_CH_3_), 1.35 (t, *J* =7.3 Hz, 3H, 6-CH_2_CH_3_), 1.11 (t, *J* =7.4 Hz, 3H, 8-S-CH_2_CH_2_CH_3_); ^13^C-NMR (100 MHz, CDCl_3_) δ 180.84, 180.78, 170.7, 158.6, 155.4, 152.7, 151.1, 147.2, 126.5, 120,8, 105.4, 32.8, 31.7, 30.2, 29.1, 20.9, 13.7, 12.1; HRMS *m*/*z* 374.1172 (Calculated for C_18_H_20_N_3_O_4_S [M + H]^+^: 374.1175); purified in column chromatography with dichloromethane: ethyl acetate: petroleum ether = 2:1:4; yield: 40%.

The 8-(butylthio)-6-ethyl-2,4-dimethylpyrimido[4,5-*c*]isoquinoline-1,3,7,10(2*H*,4*H*)-tetraone (**32**): Prepared from **3** and butane-1-thiol using general procedure B; orange solid; mp 158.8–160.5 °C; ^1^H-NMR (400 MHz, CDCl_3_) δ 6.67 (s, 1H, 9-H), 3.75 (s, 3H, 2-*N*CH_3_), 3.46 (s, 3H, 4-*N*CH_3_), 3.37 (q, *J* = 7.3 Hz, 2H, 6-CH_2_CH_3_), 2.82 (t, *J* = 7.4 Hz, 2H, 8-S-CH_2_CH_2_CH_2_CH_3_), 1.75 (dt, *J* = 15.0, *J* = 7.4 Hz, 2H, 8-S-CH_2_CH_2_CH_2_CH_3_),1.51 (dq, *J* = 14.6, *J* = 7.3 Hz, 2H, 8-CH_2_CH_2_CH_2_CH_3_), 1.34 (t, *J* = 7.3 Hz, 3H, 6-CH_2_CH_3_), 0.97 (t, *J* = 7.4 Hz, 3H, 8-CH_2_CH_2_CH_2_CH_3_); ^13^C-NMR (100 MHz, CDCl_3_) δ 180.82, 180.75, 170.7, 158.5, 155.4, 152.7, 151.1, 147.1, 126.4, 120.8, 105.4, 31.7, 30.6, 30.2, 29.3, 29.1, 22.2, 13.5, 12.1; HRMS *m*/*z* 388.1326 (Calculated for C_19_H_22_N_3_O_4_S [M + H]^+^: 388.1331); purified in column chromatography with dichloromethane: ethyl acetate: petroleum ether = 4:0.5:3; yield: 58%.

The 8-pentylthio-6-ethyl-2,4-dimethyl-pyrimido[4,5-*c*]isoquinoline-1,3,7,10(2*H*,4*H*)-tetraone (**33**): Prepared from **3** and pentane-1-thiol using general procedure B; orange solid; mp 158.6–160.4 °C; ^1^H-NMR (400 MHz, CDCl_3_) δ 6.67 (s, 1H, 9-H), 3.75 (s, 3H, 2-*N*CH_3_), 3.46 (s, 3H, 4-*N*CH_3_), 3.37 (q, *J* = 7.3 Hz, 2H, 6-CH_2_CH_3_), 2.81 (t, *J* = 7.3 Hz, 2H, 8-S-CH_2_CH_2_CH_2_CH_2_CH_3_), 1.77 (dt, *J* = 7.5, *J* = 7.4 Hz, 2H, 8-S-CH_2_CH_2_CH_2_CH_2_CH_3_), 1.46 (dt, *J* = 14.2, *J* = 6.9 Hz, 2H, 8-S-CH_2_CH_2_CH_2_CH_2_CH_3_), 1.37–1.28 (m, 5H, 6-CH_2_CH_3_ and 8-S-CH_2_CH_2_CH_2_CH_2_CH_3_), 0.97 (t, *J* = 7.4 Hz, 3H, 8-S-CH_2_CH_2_CH_2_CH_2_CH_3_); ^13^C-NMR (100 MHz, CDCl_3_) δ 180.95, 180.86, 170.81, 158.64, 155.54, 152.78, 151.23, 147.24, 126.54, 120.93, 105.51, 31.81, 31.26, 30.93, 30.27, 29.17, 27.14, 22.28, 13.99, 12.22; HRMS *m*/*z* 402.1483 (Calculated for C_20_H_24_N_3_O_4_S [M + H]^+^: 402.1488); purified in column chromatography with dichloromethane: ethyl acetate: petroleum ether = 12:1:9; yield: 43%.

The 8-hexylthio-6-ethyl-2,4-dimethylpyrimido[4,5-*c*]isoquinoline-1,3,7,10(2*H*,4*H*)-tetraone (**34**): Prepared from **3** and hexane-1-thiol using general procedure B; orange solid; mp 134.6–137.0 °C; ^1^H-NMR (400 MHz, CDCl_3_) δ 6.65 (s, 1H, 9-H), 3.73 (s, 3H, 2-*N*CH_3_), 3.44 (s, 3H, 4-*N*CH_3_), 3.35 (q, *J* = 7,3 Hz, 2H, 6-CH_2_CH_3_), 2.80 (t, *J* = 7.3 Hz, 2H, 8-S-CH_2_CH_2_CH_2_CH_2_CH_2_CH_3_), 1.74 (q, *J* = 7.3 Hz, 2H, 8-S-CH_2_CH_2_CH_2_CH_2_CH_2_CH_3_), 1.47 (q, 2H, 8-CH_2_CH_2_CH_2_CH_2_CH_2_CH_3_), 1.47 (m, 4H, 8-CH_2_CH_2_CH_2_CH_2_CH_2_CH_3_ and 8-CH_2_CH_2_CH_2_CH_2_CH_2_CH_3_), 1.32 (t, *J* = 7.3 Hz, 3H, 6-CH_2_CH_3_), 0.88 (t, *J* = 7.3 Hz, 3H, 8-CH_2_CH_2_CH_2_CH_2_CH_2_CH_3_); ^13^C-NMR (100 MHz, CDCl_3_) δ 180.8, 180.7, 170.7, 158.4, 155.4, 152.5, 151.1, 147.1, 126.4, 120.8, 105.4, 31.7 31.2, 30.8, 30.1, 29.0, 28.7, 27.3, 22.5, 14.0, 12.1; HRMS *m*/*z* 402.1483 (Calculated for C_20_H_24_N_3_O_4_S [M + H]^+^: 402.1488); purified in column chromatography with dichloromethane: ethyl acetate: petroleum ether = 2:1:4; yield: 52%.

The 6-ethyl-2,4-dimethyl-8,9-bis(phenylthio)pyrimido[4,5-*c*]isoquinoline-1,3,7,10(2*H*,4*H*)-tetraone (**35**): Prepared from **3** and benzenethiol using general procedure D; red solid; mp 188.9–191.5 °C; ^1^H-NMR (400 MHz, CDCl_3_) δ 7.58–7.54 (m, 2H, 8-C_6_H_5_ or 9-C_6_H_5_), 7.43–7.37 (m, 5H, 8-C_6_H_5_ or 9-C_6_H_5_), 7.34–7.27 (m, 3H, 8-C_6_H_5_ or 9-C_6_H_5_), 3.71 (s, 3H, 2-*N*CH_3_), 3.31 (s, 3H, 4-*N*CH_3_), 3.06 (q, *J* = 7.4 Hz, 2H, 6-CH_2_CH_3_), 1.13 (q, *J* = 7.4 Hz, 3H, 6-CH_2_CH_3_); ^13^C-NMR (100 MHz, CDCl_3_) δ 179.3, 176.8, 169.9, 157.6, 152.1, 151.1, 150.5, 147.7, 143.7, 133.3, 133.2 (2C), 131.2 (2C), 130.2, 129.4 (2C), 129.3 (2C), 128.9, 127.9, 122.1, 104.8, 31.0, 30.1, 28.8, 12.3; HRMS *m*/*z* 516.1058 (Calculated for C_27_H_22_N_3_O_4_S_2_ [M + H]^+^: 516.1052); purified in column chromatography with dichloromethane: ethyl acetate: petroleum ether = 4:1:5; yield: 39%.

The 8,9-bis(4-chlorophenylthio)-6-ethyl-2,4-dimethylpyrimido[4,5-*c*]isoquinoline-1,3,7,10(2*H*,4*H*)-tetraone (**36**): Prepared from **3** and 4-chlorobenzenethiol using general procedure D; brown solid; mp 207.8–209.8 °C; ^1^H-NMR (400 MHz, CDCl_3_) δ 7.49 (d, *J* = 8.5 Hz, 2H, 2″-H and 6″-H), 7.36 (d, *J* = 8.6 Hz, 2H, 3″-H and 5″-H), 7.35 (d, *J* = 8.7 Hz, 2H, 2′-H and 6′-H), 7.29 (d, *J* = 8.5 Hz, 2H, 3′-H and 5′-H), 3.71 (s, 3H, 2-*N*CH_3_), 3.33 (s, 3H, 4-*N*CH_3_), 3.10 (q, *J* = 7.3 Hz, 2H, 6-CH_2_CH_3_), 1.17 (t, *J* = 7.3 Hz, 3H, 6-CH_2_CH_3_); ^13^C-NMR (100 MHz, CDCl_3_) δ 179.4, 176.40, 170.1, 157.6, 152.1, 151.2, 151.0, 147.8, 142.3, 135.7, 134.8 (2C), 134.3, 132.6 (2C), 131.4, 129.7 (2C), 129.5 (2C), 128.2, 121.7, 104.8, 31.2, 30.1, 28.8, 12.3; HRMS *m*/*z* 584.0263 (Calculated for C_27_H_20_Cl_2_N_3_O_4_S_2_ [M + H]^+^: 548.0272); purified in column chromatography with dichloromethane: ethyl acetate: petroleum ether = 1:1:3; yield: 49%.

The 8,9-bis(propylthio)-6-ethyl-2,4-dimethyl-pyrimido[4,5-*c*]isoquinoline-1,3,7,10(2*H*,4*H*)-tetraone (**37**): Prepared from **3** and propane-1-thiol using general procedure D; red solid; mp 138.9–140.2 °C; ^1^H-NMR (400 MHz, CDCl_3_) δ 3.75 (s, 3H, 2-*N*CH3), 3.46 (s, 3H, 4-*N*CH3), 3.34 (q, *J* = 7.3 Hz, 2H, 6-CH_2_CH_3_), 3.27 (t, *J* = 7.2 Hz, 2H, 9-CH_2_CH_2_CH_3_), 3.08 (t, *J* = 7.3 Hz, 2H, 8-CH_2_CH_2_CH_3_), 1.76 (h, 2H, 9-CH_2_CH_2_CH_3_), 1.60 (h, 2H, 8-CH_2_CH_2_CH_3_), 1.33 (t, *J* = 7.3 Hz, 3H, 6-CH_2_CH_3_), 1.07 (t, *J* = 7.3 Hz, 3H, 9-CH_2_CH_2_CH_3_), 1.00 (t, *J* = 7.3 Hz, 3H, 8-CH_2_CH_2_CH_3_); ^13^C-NMR (100 MHz, CDCl_3_) δ 180.4, 176.6, 169.8, 158.6, 152.4, 151.8, 151.1, 148.7, 142.2, 122.0, 104.6, 36.0, 35.1, 31.2, 30.1, 28.9, 24.2, 23.9, 13.3, 13.1, 12.5; HRMS *m*/*z* 448.1365 (Calculated for C_21_H_26_N_3_O_4_S_2_ [M + H]^+^: 448.1365); purified in column chromatography with dichloromethane: ethyl acetate: petroleum ether = 5:1:14; yield: 86%.

The 8-((2-bromo-4-chlorophenyl)thio)-6-ethyl-2,4-dimethylpyrimido[4,5-*c*]isoquinoline-1,3,7,10(2*H*,4*H*)-tetraone (**38**); Prepared from **3** and 2-bromo-4-chlorobenzenethiol using general procedure B; orange solid; mp 198.4–200.2 °C; ^1^H-NMR (400 MHz, CDCl_3_) δ 7.82 (d, *J* = 1.9 Hz, 1H, 3′-H), 7.59 (d, *J* = 8.3 Hz, 1H, 6′-H), 7.44 (dd, *J* = 8.3, *J* = 1.9 Hz, 1H, 5′-H), 6.06 (s, 1H, 9-H), 3,76 (s, 3H, 2-*N*CH_3_), 3,44 (s, 3H, 4-*N*CH_3_), 3.42 (q, *J* = 7.2 Hz, 2H, 6-CH_2_CH_3_), 1.37 (t, *J* = 7.2 Hz, 3H, 6-CH_2_CH_3_); ^13^C-NMR (100 MHz, CDCl_3_) δ 181.0, 180.5, 170.8, 158.3, 153.4, 152.8, 151.0, 147.1, 138.5, 138.2, 132.3, 131.2, 129.5, 128.0, 127.3, 120.5, 105.5, 31.7, 30.2, 29.1, 12.1; HRMS *m*/*z* 519.9739 (Calculated for C_21_H_16_BrClN_3_O_4_S [M + H]^+^: 519.9733); purified in column chromatography with dichloromethane: ethyl acetate: petroleum ether = 1:5:4; yield: 70%.

The 8-((2,6-dimethoxyphenyl)thio)-6-ethyl-2,4-dimethylpyrimido[4,5-*c*]isoquinoline-1,3,7,10(2*H*,4*H*)-tetraone (**39**): Prepared from **3** and 2,6-dimethoxybenzenethiol using general procedure B; red solid; mp 223.0–223.7 °C (d); ^1^H-NMR (400 MHz, CDCl_3_) δ 7.45 (t, *J* = 8.4 Hz, 1H, 4′-H), 6.66 (d, 2H, 3′-H and 5′-H), 6.07 (s, 1H, 9-H), 3.84 (s, 6H, 2′-OCH_3_ and 6′-OCH_3_), 3.74 (s, 3H, 2-*N*CH_3_), 3.43 (s, 3H, 4-*N*CH_3_), 3.41 (q, *J* = 7.2 Hz, 2H, 6-CH_2_CH_3_), 1.36 (t, *J* = 7.3 Hz, 3H, 6-CH_2_CH_3_); ^13^C-NMR (100 MHz, CDCl_3_) δ 181.3 (2C), 170.5, 161.3 (2C), 158.6, 154.1, 152.6, 151.1, 147.5, 133.2, 126.8, 121.0, 105.4, 104.5 (2C), 102.0, 56.4 (2C), 31.7, 30.2, 29.0, 12.2; HRMS *m*/*z* 468.1226 (Calculated for C_23_H_22_N_3_O_6_S [M + H]^+^: 468.1229); purified in column chromatography with dichloromethane: ethyl acetate: petroleum ether = 9:1:3; yield: 63%.

The 8-((5-bromo-2-methoxyphenyl)thio)-6-ethyl-2,4-dimethylpyrimido[4,5-*c*]isoquinoline-1,3,7,10(2*H*,4*H*)-tetraone (**40**): Prepared from **3** and 5-bromo-2-methoxybenzenethiol using general procedure B; orange solid; mp 221.0–222.0 °C (d); ^1^H-NMR (400 MHz, CDCl_3_) δ 7.63 (q, *J* = 2.4 Hz, 1H, 6′-H), 7.61 (d, *J* = 8.7, *J* = 2.6 Hz, 1H, 4′-H), 6.93 (d, *J* = 8.6 Hz, 1H, 3′-H), 6.11 (s, 1H, 9-H), 3.85 (s, 3H, 2′-OCH_3_), 3.75 (s, 3H, 2-*N*CH_3_), 3.44 (s, 3H, 4-*N*CH_3_), 3.41 (q, *J* = 7.2 Hz, 2H, 6-CH_2_CH_3_), 1.36 (t, *J* = 7.3 Hz, 3H, 6-CH_2_CH_3_); ^13^C-NMR (100 MHz, CDCl_3_) δ 181.2, 180.8, 170.7, 159.2, 158.4, 153.8, 152.7, 151.1, 147.2, 139.5, 135.7, 127.7, 120.7, 116.7, 113.5, 113.3, 105.4, 56.4, 31.7, 30.2, 29.1, 12.1; HRMS *m*/*z* 516.0237 (Calculated for C_22_H_19_BrN_3_O_5_S [M + H]^+^: 516.0229); purified in column chromatography with dichloromethane: ethyl acetate: petroleum ether = 20:1:4; yield: 76%.

The 8-((3,5-dichlorophenyl)thio)-6-ethyl-2,4-dimethylpyrimido[4,5-*c*]isoquinoline-1,3,7,10(2*H*,4*H*)-tetraone (**41**): Prepared from **3** and 3,5-dichlorobenzenethiol using general procedure B; yellow solid; mp 179.8–182.0 °C; ^1^H-NMR (400 MHz, CDCl_3_) δ 7.44 (s, 2H, 2′-H and 6′-H), 7.52 (s, 1H, 4′-H); 6.24 (s, 1H, 9-H), 3.75 (s, 3H, 2-*N*CH_3_), 3.43 (s, 3H, 4-*N*CH_3_), 3.40 (q, *J* = 7.3 Hz, 2H, 6-CH_2_CH_3_), 1.36 (t, *J* = 7.3 Hz, 3H, 6-CH_2_CH_3_); ^13^C-NMR (100 MHz, CDCl_3_) δ 181.1, 180.3, 170.9, 158.6, 145.9, 152.8, 151.0, 147.0, 136.6, 133.6 (2C), 131.1, 130.5, 128.4 (2C), 120.3, 105.5, 31.7, 30.2, 29.1, 12.1; HRMS *m*/*z* 476.0235 (Calculated for C_21_H_16_Cl_2_N_3_O_4_S [M + H]^+^: 476.0239); purified in column chromatography with dichloromethane: ethyl acetate: petroleum ether = 20:1:7; yield: 69%.

The 8-phenylthio-2,4,6-trimethyl-pyrimido[4,5-*c*]isoquinoline-1,3,7,10(2*H*,4*H*)-tetraone (**42**): Prepared from **4** and benzenethiol using general procedure B; yellow solid; mp 206.0–208.0 °C (d); ^1^H-NMR (400 MHz, CDCl_3_) δ 7.53 (m. 5H, 2′-H, 3′-H, 4′-H, 5′-H and 6′-H), 6.18 (s, 1H, 9-H), 3.74 (s, 3H, 2-*N*CH_3_), 3.43 (s, 3H, 4-*N*CH_3_), 3.01 (s, 3H, 6-CH_3_); ^13^C-NMR (100 MHz, CDCl_3_) δ 181.1, 180.9, 166.3, 158.3, 156.5, 152.7, 151.0, 146.9, 135.7 (2C), 130.7, 130.5 (2C), 128.1, 127.1, 120.9, 105.7, 30.2, 29.1, 26.9; HRMS *m*/*z* 394.0862 (Calculated for C_20_H_16_N_3_O_4_S [M + H]^+^: 394.0862); purified in column chromatography with dichloromethane: ethyl acetate: petroleum ether = 12:1:8; yield: 65%.

The 8-(4-methoxy-phenylthio)-2,4,6-trimethylpyrimido[4,5-*c*]isoquinoline-1,3,7,10(2*H*,4*H*)-tetraone (**43**): Prepared from **4** and 4-methoxybenzenethiol using general procedure B; orange solid; mp 198.0–199.0 °C (d); ^1^H-NMR (400 MHz, CDCl_3_) δ 7.43 (d. 2H, 3′-H and 5′-H), 7.02 (d. 2H, 2′-H and 6′-H), 6.16 (s, 1H, 9-H), 3.86 (s, 3H, 4′-OCH_3_), 3.73 (s, 3H, 2-*N*CH_3_), 3.43 (s, 3H, 4-*N*CH_3_), 3.00 (s, 3H, 6-CH_3_); ^13^C-NMR (100 MHz, CDCl_3_) δ 181.2, 181.0, 166.2, 161.6, 158.3, 157.3, 152.6, 151.0, 147.0, 137.2 (2C), 128.0, 121.0, 117.2, 116.1 (2C), 105.7, 55.5, 30.2, 29.0, 26.9; HRMS *m*/*z* 424.0963 (Calculated for C_21_H_18_N_3_O_5_S [M + H]^+^: 424.0967); purified in column chromatography with dichloromethane: ethyl acetate: petroleum ether = 3:1:4; yield: 69%.

The 8-((4-fluorophenyl)thio)-2,4,6-trimethylpyrimido[4,5-*c*]isoquinoline-1,3,7,10(2*H*,4*H*)-tetraone (**44**): Prepared from **4** and 4-fluorobenzenethiol using general procedure B; yellow solid; mp 211.0–212.0 °C (d); ^1^H-NMR (400 MHz, CDCl_3_) δ 7.52 (dd, *J*_H,H_ = 8.7, *J*_H,H_ = 5.2 Hz, 2H, 2′-H and 6′-H), 7.21 (t, *J*_H,H_ = 8.5, *J*_F,H_ = 8.5 Hz, 2H, 3′-H and 5′-H), 6.15 (s, 1H, 9-H), 3.73 (s, 3H, 2-*N*CH_3_), 3.43 (s, 3H, 4-*N*CH_3_), 3.01 (s, 3H, 6-CH_3_); ^13^C-NMR (100 MHz, CDCl_3_) δ 181.2, 180.9, 166.4, 164.4 (d, 1C, *J*_F,C_ = 252.9 Hz, 4′), 158.4, 156.4, 152.8, 151.1, 146.9, 138.0 (d, 2C, *J*_F,C_ = 8.8 Hz, 2′ and 6′), 128.2, 122.5 (d, 1C, *J*_F,C_ = 3.6 Hz, 1′), 121.0, 118.0 (d, 2C, *J*_F,C_ = 22.1 Hz, 3′ and 5′), 105.8, 30.4, 29.2, 27.0; HRMS *m*/*z* 412.0771 (Calculated for C_20_H_15_FN_3_O_4_S [M + H]^+^: 412.0767); purified in column chromatography with dichloromethane: ethyl acetate: petroleum ether = 1:1:2; yield: 69%.

### 5.3. Crystallography

#### 5.3.1. Preparation of Single Crystals

Single crystals were grown by solvent evaporation at room temperature from the synthesis products. Crystals suitable for X-ray diffraction studies were obtained from crystallization in saturated solutions: tetrahydropyran for **7** and benzene for **16**.

#### 5.3.2. Single Crystal X-ray Diffraction

Crystals were prepared under inert conditions and immersed in perfluoropolyether as a protective oil for manipulation. Suitable crystals were mounted on MiTeGen Micromounts^TM^, and these samples were used for data collection. Data were collected with a D8 Venture diffractometer CuKα, 298 K (Bruker, Karlsruhe, Germany). The data were processed with the APEX3 program [22] and corrected for absorption using SADABS [23]. The structures were resolved using direct methods [24], which revealed the position of all nonhydrogen atoms. These atoms were refined on F2 by a full-matrix least-squares procedure using anisotropic displacement parameters. All hydrogen atoms were located in different Fourier maps and were included as fixed contributions riding on attached atoms with isotropic thermal displacement parameters 1.2 (C–H) or 1.5 (methyl) times those of the respective atom. CCDC 2023040 (Compound **38**), CCDC 2023047 (Compound **30**), CCDC 2023048 (Compound **27**), CCDC 2023049 (Compound **28**), CCDC 2023045 (Compound **P7**), CCDC 2023064 (Compound **32**), and CCDC 2023061 (Compound **23**) contain the crystallographic data listed in Appendix A. These data can be obtained free of charge from the Cambridge Crystallographic Data Centre.

### 5.4. Images of 3D-Models

We processed with the BIOVIA Discovery Studio Visualizer software [25] from the .cif files. The 3D models were superimposed considering the quinone tricyclic core as temperate. The dihedral angle was measured by the same software. The images were created with the script/Visualization/Publication Quality.

### 5.5. Evaluation of Antibacterial Activity

Antimicrobial activity in vitro against *Staphylococcus aureus* methicillin-susceptible strain (ATCC^®^ 29213), *Staphylococcus aureus* methicillin-susceptible strain (ATCC^®^ 43300), *Enterococcus faecalis* (ATCC^®^ 29212), *Escherichia coli* (ATCC^®^ 25922), *Pseudomonas aeruginosa* (ATCC^®^ 25923), and *Klebsiella pneumoniae* (ATCC^®^ 700603) were investigated by minimum inhibitory concentration (MIC) of a broth microdilution method, according to recommendations of the Clinical and Laboratory Standards Institute (CLSI) [26]. All compounds tested were dissolved in dimethyl sulfoxide (DMSO) to levels not exceeding 1% per well. Vancomycin and gentamicin were used as references against the strains, and the results were compared to the MIC ranges reported by the CLSI as a quality control measure [19]. In addition, one well in each plate with medium without antibiotics was used as a positive control for bacterial growth. We also used a well containing only medium without the bacterial inoculum as a sterility control of the procedure. The compounds were tested from the maximum concentration reached and standard drugs since 16 μg/mL. The inoculum was prepared to a turbidity equivalent of an 0.5 McFarland standard, diluted in broth media to give a final concentration of 5 × 10^5^ CFU/mL in the test tray; they were covered and placed in plastic bags to prevent evaporation. The plates were incubated at 35 °C for 18–20 h. The MIC was defined as the lowest concentration of the compound giving complete inhibition of visible growth. All experiments were performed three times in triplicate.

## 6. Patents

PatentWO2017113031A1, PCT/CL2015003780A1, USAUS11390622B2, EPO EP3404026A4; China CN109121411B. MX/a/2018/008192A titled: “Pyrimidine-Isoquinoline-Quinone Derived Compounds, their Salts, Isomers, Pharmaceutically Acceptable Tautomers; Pharmaceutical Composition; Preparation Procedure; and their Use in the Treatment of Bacterial and Multi-Resistant Bacterial Diseases”.

## Figures and Tables

**Figure 1 antibiotics-12-01065-f001:**
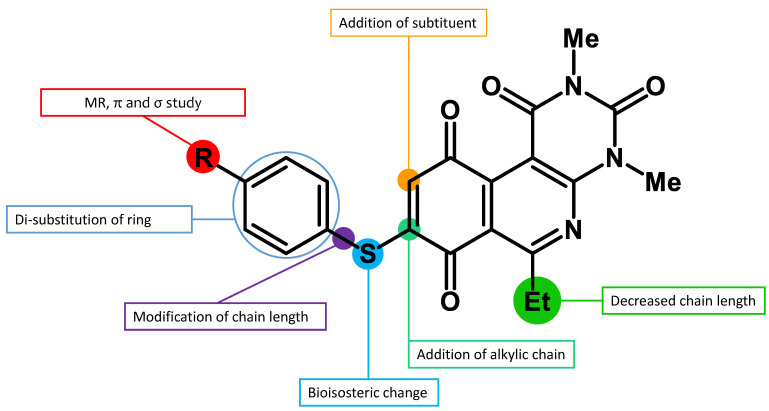
A summary of the structural modifications to the SAR study.

**Figure 2 antibiotics-12-01065-f002:**
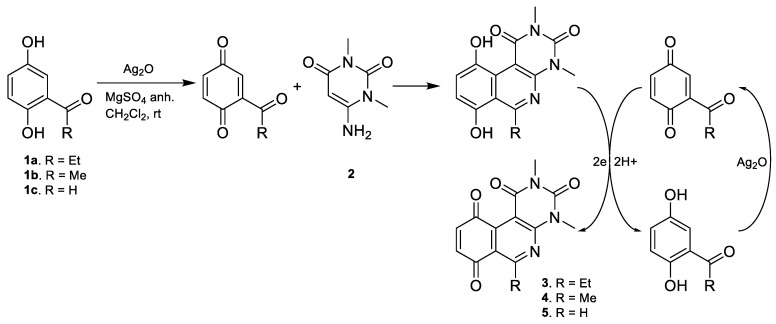
Reaction mechanism proposed for obtaining quinone tricyclic core by one-pot reaction.

**Figure 3 antibiotics-12-01065-f003:**
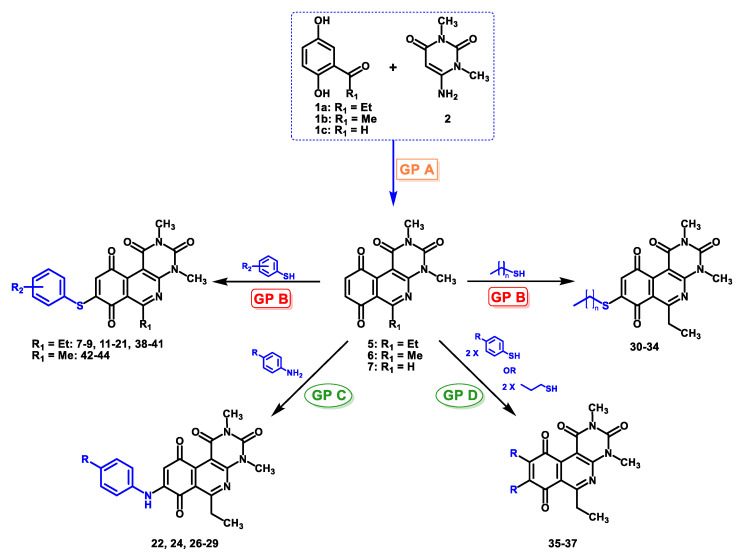
General scheme for obtaining target compounds through the general procedures used.

**Figure 4 antibiotics-12-01065-f004:**
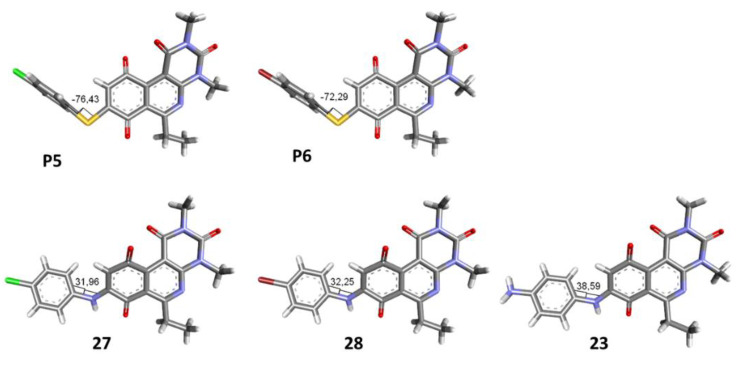
The 3D-dimensional models obtained by X-ray diffraction of individual crystals of compounds **P5**, **P6**, **23**, **27,** and **28**. The dihedral angle of each compound is shown.

**Figure 5 antibiotics-12-01065-f005:**
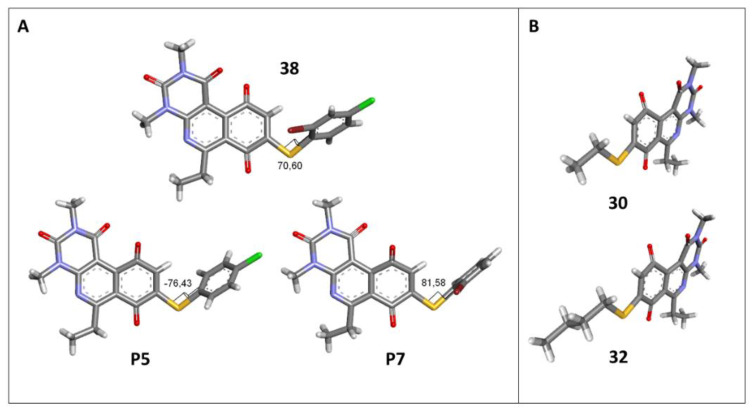
Three-dimensional models obtained by X-ray diffraction of individual crystals of compounds **30**, **32**, **38**, **P5,** and **P7**. (**A**) Comparison of dihedral angles among derived thiophenolic compounds; (**B**) Comparison of geometry of alkylthiol derivatives.

**Table 2 antibiotics-12-01065-t002:** Summary of the Free-Wilson coefficients analysis.

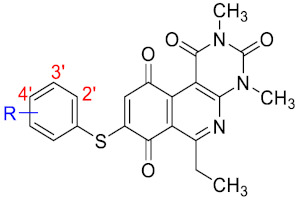	Coef. FW					
	R	Me	MeO	F	Cl	Br
	2′ (*ortho*)	−0.9653	0.255	0	0	0.602
	3′ (*meta*)	−0.0622	−0.046	−0.0581	0.2594	0.301
	4′ (*para*)	−0.0622	−0.6481	−0.3591	−0.0416	0
	Intercept: 5.0849					

**Table 3 antibiotics-12-01065-t003:** Compounds designed from the Free-Wilson analysis.

Compound	38	39	40	41
Structure	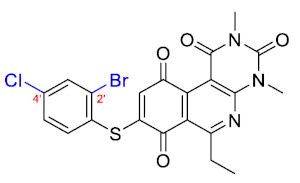	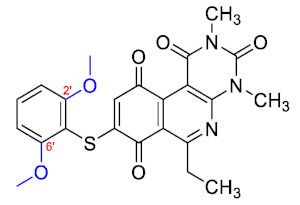	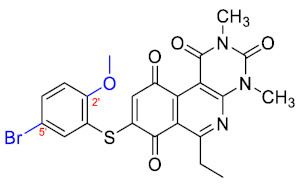	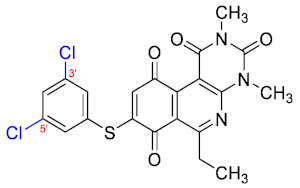
FWc	2′-Br = 0.602 4′-Cl = −0.0416	2′-OMe = 0.255	2′-OMe = 0.255 3′-Br = 0.301	3′-Cl = 0.2594
FWc = Free-Wilson coefficient.				

## Data Availability

Thesis report for the degree of Pharmaceutical Chemist of A.P.-R. https://bibliotecadigital.uchile.cl/permalink/56UDC_INST/llitqr/alma991002956919703936 (accessed on 31 May 2023); Doctoral thesis report of J.A.-L. Link not available due to restricted circulation period.

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
