# Peer review of "Design, Synthesis, and Structure–Activity Relationship Studies of New Quinone Derivatives as Antibacterial Agents"

_antibiotics, 2023, doi:10.3390/antibiotics12061065_

Round 1
Reviewer 1 Report
This study illustrates the development of a new family of antibacterial agents called pyrimidoisoquinolinequinones. A new family of pyrimidoisoquinolinequinones was designed to target multidrug-resistant gram-positive bacteria. The authors have found that the compounds with the bridging sulfur atom, bearing a para-substituted benzene ring and lipophilicity, exhibit the highest antimicrobial activity. The replacement of an ethyl group by a methyl group at C-6 position of the quinone tricyclic core also generates active compounds with similar activity. However, the introduction of carbons between the sulfur atom and the benzene ring, as well as the replacement of the aromatic ring by an alkyl chain, annuls the activity of the compounds. Homo disubstitution of thiophenolic or thioalkyl groups on the quinone core generated inactive compounds.
The bioisosteric replacement of the sulfur atom by one of nitrogen produced a change in the geometry, by reducing the dihedral angle between the substituted benzene ring and the quinone nucleus. This change increased the activity when the benzene ring presented an electron attracting atom such as chlorine or bromine, on the other hand, the presence of electron donor groups such as amines lowered the activity.
The authors concluded that this new family of compounds displayed a high potential for improvement in their performance as potent antimicrobial drugs against Gram-positive bacteria. Finally, it is important to highlight the bioactivity of compounds 11 and 41, which open the way for studying new active structures on K. pneumoniae, a pathogen considered a priority by the WHO.
These compounds are more effective against Gram-positive bacteria when they have the following properties: a bridging sulfur atom, a para-substituted benzene ring, lipophilicity, a methyl group at the C-6 position of the quinone tricyclic core, and an electro-attracting atom such as chlorine or bromine on the benzene ring.
The authors believe that these compounds have the potential to be developed into new drugs that can treat infections caused by Gram-positive bacteria.
The authors have written a well-structured and informative manuscript that makes a significant contribution to the field. The study is well-designed and executed, and the results are promising.
Reviewer 2 Report
Overall Comment on the Article:
The manuscript entitled "Design, synthesis, and structure-activity relationship studies of new quinone derivatives as antibacterial agents" is an excellent contribution to the field of medicinal chemistry and an alarming global health problem – resistance of microorganisms to antibiotics. The authors have collected and presented comprehensive data on a series of new quinone derivatives, totaling 40 chemical entities. The use of NMR and crystallography has allowed for the accurate characterization and identification of these chemical entities. The authors have created a synthetic library of compounds that will undoubtedly make progress in the identification of new antibiotics. Presented results showed that this new family of compounds displayed a high potential for improvement in their performance as potent antimicrobial drugs against Gram-positive bacteria strains. In summary, the paper is well written, the data presented is thorough and the authors' findings are significant contributions to the field. I recommend accepting the paper after minor revisions:
1) Lines 2 and 5: Title and period after authors are not necessary.
2) Line 90: Please add the reference.
3) Please delete Table 1. This Table contains the data from the previous report and doesn’t acquire something new.
4) Figure 2: MgSO4 should be MgSO4
5) Line 200: E. Faecalis should be E. faecalis and line 203 K. pneumoniae should be in italics.
6) The study of the structure-activity relationship established that: Comment It is not necessary, please delete or revise.
Minor editing of English language required.
Reviewer 3 Report
I have evaluated the manuscript (Antibiotics-2456008) titled “Design, synthesis, and structure-activity relationship studies of new quinone derivatives as antibacterial agents” by Vásquez-Velásquez and coworkers evaluated the activity of new quinone based antibacterial agents against multidrug-resistant gram-positive bacteria. I found this article interesting for the readers and followed the journal Antibiotics’ scope. However, to improve the standard of this manuscript some part of the discussion of chemistry should move to the supplementary, and author should discuss the significant finding (move some compounds to the supplementary) in the manuscript and move others to the supplementary.
I would recommend this article be published in Antibiotics after minor corrections.
The author needs to address the following comments/corrections.
1. Supplementary should contain table of content with author’s name.
2. The author could mention the number of antibiotics approved in last ten years in the abstract instead of mentioning few antibacterial drugs have been approved by the FDA in the last 10 years.
3. The introduction should be concise, and more focus should be on the present work highlighting the background of the research. Some part of the introduction should move to result and discussion.
4. The author should move the bulk of experimental part to the supplementary.
5. In the supplementary show each reaction depicted in the scheme 3 separately.
6. Authors should provide NMR spectra of all new compounds which should include peak picking.
7. All the optimization of reaction condition for all reactions should be included in the supplementary in the tabular format.
8. The author should mention the control (positive or negative) used for antimicrobial activity in the table.
9. The author should show the diagram of Crystal structure in the supplementary.
10. The author follows the correct format of references (e.g Year Bold, Volume Italic etc).
11. The author could include the following relevant references.
(a) https://doi.org/10.1021/jm801241n
(b) https://doi.org/10.1016/j.bmcl.2008.05.004
